# GeoDM: Geometry-aware Distribution Matching for Dataset Distillation

## Abstract

Dataset distillation aims to synthesize a compact subset of the original data, enabling models trained on it to achieve performance comparable to those trained on the original large dataset. Existing distribution-matching methods are confined to Euclidean spaces, making them only capture linear structures and overlook the intrinsic geometry of real data, e.g., curvature. However, high-dimensional data often lie on low-dimensional manifolds, suggesting that dataset distillation should have the distilled data manifold aligned with the original data manifold. In this work, we propose a geometry-aware distribution-matching framework, called **GeoDM**, which operates in the Cartesian product of Euclidean, hyperbolic, and spherical manifolds, with flat, hierarchical, and cyclical structures all captured by a unified representation. To adapt to the underlying data geometry, we introduce learnable curvature and weight parameters for three kinds of geometries. At the same time, we design an optimal transport loss to enhance the distribution fidelity. Our theoretical analysis shows that the geometry-aware distribution matching in a product space yields a smaller generalization error bound than the Euclidean counterparts. Extensive experiments conducted on standard benchmarks demonstrate that our algorithm outperforms state-of-the-art data distillation methods and remains effective across various distribution-matching strategies for the single geometries.

## 1 Introduction

Dataset distillation (DD) (Wang et al., 2018) aims to reduce the storage and computational burden of large-scale datasets by synthesizing a compact set of informative samples. In addition to optimization-based approaches such as gradient matching (Zhao et al., 2020a) and trajectory matching (Cazenavette et al., 2022), distribution matching (DM) (Zhao & Bilen, 2023) has gained particular traction due to its efficiency and conceptual simplicity, as it aligns the distributions of real and synthetic data rather than tracking optimization dynamics. A variety of DM methods have since been proposed: M3D (Zhang et al., 2024) strengthens distribution-level consistency via higher-order moments, DSDM (Li et al., 2024) scales feature-level optimization, WDDD (Liu et al., 2023a) leverages Wasserstein metrics, OPTICAL (Cui et al., 2025) incorporates optimal transport, and NCFM (Wang et al., 2025) captures discrepancies by modeling both frequency and magnitude components (see Appendix F for more related work).

Despite these advances, nearly all DM-based methods conduct the matching in Euclidean latent embedding spaces (Li et al., 2025). The underlying assumption for the embedding space, albeit effective on Euclidean data, neglects curvature and non-Euclidean structures inherent in real data, which inevitably limits the capacity of distilled samples to capture underlying geometries. The embeddings in Figure 1 exhibit distinct geometries: a tree-like hierarchy characterized by the hyperbolic space (Nickel & Kiela, 2017; Ganea et al., 2018), or a shell lying in the spherical space (Davidson et al., 2018), whereas the Euclidean embedding cannot encode the hyperbolic or spherical geometries. This means that real data with hyperbolic and/or spherical geometries cannot be well captured by flat embeddings. This phenomenon resonates with the manifold hypothesis (Hinton & Salakhutdinov, 2006; Tenenbaum et al., 2000) that, high-dimensional data often lie on low-dimensional manifolds which faithfully describes data distribution. This motivates us to distill data via embedding into the product of low-dimensional manifolds where hierarchical, cyclical, or directional patterns present in the original dataset can be modeled.

In this work, we propose a geometry-aware data distillation framework which performs distribution matching in a product Riemannian space that combines Euclidean, hyperbolic, and spherical manifolds to capture intrinsic geometries of real data. Specifically, we implement this product manifold with a Riemannian convolutional neural network (Masci et al., 2015) which can handle factor-wise feature maps and geometry-aware operations. As we will see in Figure 3, this geometry-aware modeling can achieve consistently strong performance across diverse datasets and distribution matching methods. To align with the low-dimensional manifold of each dataset, we introduce learnable curvature for non-Euclidean factors and learnable weights for different geometries. The former forces the product manifold to be aligned with real data manifold, while the latter captures Euclidean, hyperbolic, and spherical contributions. They shape the embedding space such that it increasingly conforms to the geometry of the data manifold during training. Also, we design a geometry-aware optimal transport (OT) loss that measures how real and synthetic samples align across the three components of the product space. The loss couples different geometries, preserves class-conditional mass, and prevents degenerate solutions where one component dominates. In a word, distribution matching is cast into the alignment between product manifolds with learnable curvature and weights for approximate manifold learning while maintaining the efficiency of modern dataset distillation.

Furthermore, we introduce theoretical foundations for the geometry-aware dataset distillation. Under mild regularity assumptions (Assumption 4.1), a high-dimensional sample $x$ can be decomposed into three parts: content about task-relevant information, noise describing stochastic fluctuations, and geometry of the underlying manifold. While most existing approaches implicitly merge the geometry part into the content part or simply ignore it, our analysis in Theorem 4.1 shows that the geometry part plays a crucial role in preserving distributional fidelity during distillation, and that in contrast, Euclidean embeddings tend to suppress or flatten this signal. Given this finding, Theorem 4.2 proves that distribution matching in a latent product space, that combines Euclidean, hyperbolic, and spherical components, yields a strictly tighter generalization error bound than in single Euclidean latent spaces. This implies that the way we do distribution matching here can not only approximate manifold learning itself but also guide synthetic data to respect structural constraints of real data distributions.

To summarize, we make the following contributions in this work:

- We build on the manifold hypothesis to consider dataset distillation as a problem of distribution matching between product manifolds for embedding real and synthetic data, where each factor space comes with a unique curvature information, e.g., Euclidean, or hyperbolic, spherical.

- We design **GeoDM**, a distribution-matching framework operating in a Cartesian product of Euclidean, hyperbolic, and spherical spaces, where learnable curvature and weights are introduced to capture the intrinsic geometry of real data, and an optimal transport loss is designed to enhance distributional alignment.

- We conduct the theoretical analysis which shows that the geometric structure of data plays a crucial role in dataset distillation, and performing distribution matching provides a tighter generalization error bound in Cartesian product spaces than in single Euclidean spaces.

- We conduct extensive experiments on standard benchmarks which show that our method consistently surpasses state-of-the-art dataset distillation algorithms and remains robust across different distribution-matching strategies for individual geometries.

## 2 PRELIMINARIES

### 2.1 DATASET DISTILLATION

Dataset distillation (DD) seeks to synthesize a smaller dataset $\mathcal{S} = \{(\tilde{x}_i, \tilde{y}_i)\}_{i=1}^{m}$ that preserves the essential information of a large dataset $\mathcal{D} = \{(x_i, y_i)\}_{i=1}^{n}$, where $n \gg m$. Formally, the distilled data set is obtained by

$$\mathcal{S}^{\star} = \arg\min_{\mathcal{S}} D\big(\phi(\mathcal{D}), \phi(\mathcal{S})\big),$$

where $D(\cdot, \cdot)$ denotes a divergence (e.g., maximum mean discrepancy, or Wasserstein distance) between representations computed by a feature map $\phi$. The distilled dataset $\mathcal{S}^*$ is then used to train a model whose performance approaches that of the model trained on the full data $\mathcal{D}$.

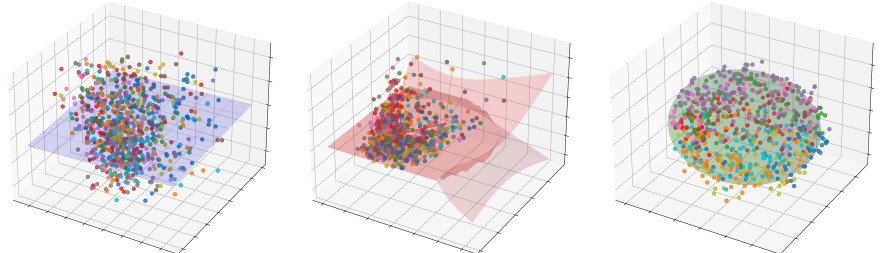

Figure 1: 3D embeddings of CIFAR10 data in Euclidean (left), hyperbolic (middle), and spherical (right) spaces with fitted manifolds: in hyperbolic space the points exhibit a hierarchical pattern and align well with the hyperbolic surface, while in spherical space they concentrate on a sphere, revealing inherent spherical geometry; the Euclidean space struggles to capture such geometric structure.

Among DD methods, DM stands out for its efficiency and avoidance of complex bi-level optimization. DM focuses on aligning the empirical distributions of real and synthetic data in an embedding space. Let $\psi_\vartheta : \mathbb{R}^d \to \mathbb{R}^{d'}$ be an embedding function parameterized by $\vartheta \sim P_\vartheta$ (e.g., a randomly initialized neural network), and Let $A(\cdot, \omega)$ be a differentiable data augmentation operator parameterized by $\omega \sim \Omega$, where $\Omega$ denotes the domain (or distribution) of augmentation parameters, referred to the generalization ball. The DM objective is:

$$\min_{\mathcal{S}} \ \mathbb{E}_{\vartheta \sim P_\vartheta, \omega \sim \Omega} \left\| \frac{1}{|\mathcal{D}|} \sum_{(x,y) \in \mathcal{D}} \psi_\vartheta\left(A(x, \omega)\right) - \frac{1}{|\mathcal{S}|} \sum_{(\tilde{x}, \tilde{y}) \in \mathcal{S}} \psi_\vartheta\left(A(\tilde{x}, \omega)\right) \right\|_2.$$

By averaging over multiple embeddings $\psi_\vartheta$, DM provides a robust approximation of the input space, enabling direct optimization of $\mathcal{S}$ without tracking full training dynamics.

However, traditional DM methods operate exclusively in Euclidean latent spaces, where DM is limited to Euclidean data. This motivates us to extend DM to product spaces that incorporate diverse Riemannian geometries to align better with the underlying manifold structures of real data.

## 2.2 Riemannian Manifolds

Real-world data often exhibit non-linear structures that are poorly captured by Euclidean geometry alone, motivating the use of Riemannian manifolds to model curved spaces (Do Carmo & Flaherty Francis, 1992). A Riemannian manifold $(\mathcal{M}, g)$ is a smooth space equipped with a metric $g$ that defines distances and angles locally. We focus on three families of constant-curvature manifolds: Euclidean (zero curvature), hyperbolic (negative curvature), and spherical (positive curvature). These spaces capture flat, hierarchical, and cyclical patterns, respectively.

The Euclidean space $\mathbb{E}^n$ with dimension $n$ is the familiar flat geometry with the standard inner product metric $g^{\mathbb{E}}$. It excels at modeling Euclidean data but struggles with non-Euclidean structure.

The hyperbolic space $\mathbb{H}_c^n$ with curvature $c < 0$ is well-suited for hierarchical data due to its exponential volume growth. Under the Poincaré ball model (Mathieu et al., 2019), $\mathbb{H}_c^n = \left\{ z \in \mathbb{R}^n : \|z\|_2 < \frac{1}{\sqrt{-c}} \right\}$ with metric $g_z^{\mathbb{H}} = \lambda_z^2 \, g^{\mathbb{E}}$ for $\lambda_z = \frac{2}{1 - c\|z\|_2^2}$, where $z$ is the embedding.

The spherical space $\mathbb{S}_c^n$ with curvature $c > 0$ models directional or cyclical data, such as angles or cycles. It is defined as $\mathbb{S}_c^n = \{ z \in \mathbb{R}^{n+1} : \langle z, z \rangle = 1/c \}$, with metric $g_z^{\mathbb{S}}$ induced from the ambient Euclidean space, preserving rotational symmetries (Meilă & Zhang, 2023).

To handle mixed structures in real data, the following product manifold can be constructed:

$$\mathcal{M} = \mathcal{M}_1 \times \mathcal{M}_2 \times \cdots \times \mathcal{M}_K,$$

with metric $g^{\mathcal{M}} = (g^{\mathcal{M}_1}, \cdots, g^{\mathcal{M}_K})$, where each $\mathcal{M}_k$ is $\mathbb{E}^{d_k}$, $\mathbb{H}_{c_k}^{d_k}$, or $\mathbb{S}_{c_k}^{d_k}$. For any point $z = (z^{(1)}, \ldots, z^{(K)}) \in \mathcal{M}$, the tangent space decomposes additively, and distances are computed component-wise. This product construction allows us to embed data into a unified space that adapts to diverse geometries, forming the basis for our geometry-aware distribution matching in GeoDM.

## 3 PROPOSED METHOD - GEODM

We now develop a geometry-aware data distillation framework which can take advantage of the Euclidean–hyperbolic–spherical product manifold to approximate the underlying data manifold. The training procedure of our framework is illustrated in Figure 2, with pseudo code described by Algorithm 1 in Appendix C.

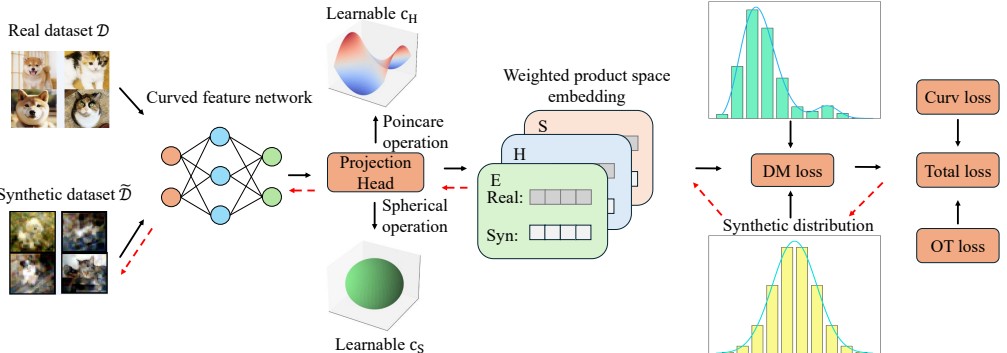

Figure 2: Overall pipeline of our GeoDM: Real and synthetic data are first processed by a curved feature network, where hyperbolic and spherical branches incorporate learnable curvature and dedicated projection heads. The resulting embeddings are mapped to the Euclidean–hyperbolic–spherical product space. DM aligns real and synthetic features, while a geometry-aware optimal transport (OT) loss further couples the three geometries and preserves class-conditional mass.

### 3.1 PRODUCT SPACE FOR DISTRIBUTION MATCHING

A central part of our framework is the design of the *product space* that admits multiple geometries. Specifically, we define

$$\mathcal{P} = \mathbb{E}^{d_E} \times \mathbb{H}^{d_H}_{c_H} \times \mathbb{S}^{d_S}_{c_S}, \tag{1}$$

where $\mathbb{E}^{d_E}$ is the $d_E$-dimensional Euclidean space, $\mathbb{H}^{d_H}_{c_H}$ is the $d_H$-dimensional hyperbolic space with curvature $c_H < 0$, and $\mathbb{S}^{d_S}_{c_S}$ is the $d_S$-dimensional spherical space with curvature $c_S > 0$. Each real or synthetic sample is embedded into $\mathcal{P}$, where three components stands for complementary geometric properties of the data.

To embed data into $\mathcal{P}$, we implement a Riemannian convolutional neural network (CNN) (Masci et al., 2015), which generalizes convolutional operations to non-Euclidean domains. In particular, spherical convolutions are used to extract features invariant to rotations on $\mathbb{S}^{d_S}_{c_S}$, while hyperbolic layers map Euclidean features into $\mathbb{H}^{d_H}_{c_H}$ through exponential and logarithmic maps to ensure consistency with the hyperbolic geometry. Together with standard Euclidean convolutions, this design yields geometry-aware features that align with their respective factor spaces of $\mathcal{P}$.

Given real and synthetic embeddings $z_i = (z_i^E, z_i^H, z_i^S)$, $\tilde{z}_j = (\tilde{z}_j^E, \tilde{z}_j^H, \tilde{z}_j^S) \in \mathcal{P}$, we form a product-space feature by concatenating the three geometric components:

$$\Psi_{\mathcal{P}}(z) = \left[ \psi_E(z^E) \; ; \; \psi_H(z^H) \; ; \; \psi_S(z^S) \right] \quad \text{and} \quad \Psi_{\mathcal{P}}(\tilde{z}) = \left[ \psi_E(\tilde{z}^E) \; ; \; \psi_H(\tilde{z}^H) \; ; \; \psi_S(\tilde{z}^S) \right].$$

We then align the real and synthetic product features via the NCFM objective (Wang et al., 2025):

$$\mathcal{L}_{\text{DM}} = \text{NCFM}\left( \{\Psi_{\mathcal{P}}(z_i)\}_{i=1}^n, \; \{\Psi_{\mathcal{P}}(\tilde{z}_j)\}_{j=1}^m \right), \tag{2}$$

which performs distribution matching directly in the product space (details of NCFM omitted here for brevity; see Wang et al. (2025)).

To further enhance the expressiveness of the product space, we introduce two complementary mechanisms, i.e., *learnable curvature* and *learnable weights*, which allow the geometry of each factor space to adapt to the intrinsic structure of relevant data and control their relative contributions during training.

**Learnable curvature.** We treat the curvature parameters of the hyperbolic and spherical components, $c_H < 0$ and $c_S > 0$, as learnable variables optimized together with the network weights. To ensure stability and prevent degenerate solutions, we introduce a curvature loss

$$\mathcal{L}_{\text{curv}} = \lambda_H \, R_H(z^H, c_H) + \lambda_S \, R_S(z^S, c_S), \tag{3}$$

where $\lambda_H, \lambda_S$ are fixed hyperparameters. Let $r_H = 1/\sqrt{-c_H}$ and $r_S = 1/\sqrt{c_S}$ denote the radius of the hyperbolic Poincaré ball and the sphere, respectively. We regularize embeddings by penalizing their deviation from these radii:

$$R_H(z^H, c_H) = \mathbb{E}\big[\,\big|\,\|z^H\|_2 - r_H\,\big|\,\big], \qquad R_S(z^S, c_S) = \mathbb{E}\big[(\|z^S\|_2 - r_S)^2\big].$$

where $R_H$ penalizes hyperbolic embeddings that approach the Poincaré-ball boundary and regularizes the magnitude of $c_H$, while $R_S$ encourages spherical embeddings to stay near the unit sphere and keeps $c_S$ well-scaled. We fix the dimensions $(d_E, d_H, d_S)$ across all datasets rather than learning them, as varying dimensionality often introduces extra degrees of freedom that destabilize optimization and reduce generalization.

**Learnable weights.** Given an embedded pair of points $z = (z^E, z^H, z^S)$ and $\tilde{z} = (\tilde{z}^E, \tilde{z}^H, \tilde{z}^S)$, the product-space distance is defined as

$$d_{\mathcal{P}}^2(z, \tilde{z}) = \alpha \, \|z^E - \tilde{z}^E\|^2 + \beta \, d_{\mathbb{H}_{c_H}}^2(z^H, \tilde{z}^H) + \gamma \, d_{\mathbb{S}_{c_S}}^2(z^S, \tilde{z}^S), \tag{4}$$

where $\alpha, \beta, \gamma > 0$ are learnable weights controlling the relative contributions of Euclidean, hyperbolic, and spherical components. We normalize them as

$$(\tilde{\alpha}, \tilde{\beta}, \tilde{\gamma}) = \text{softmax}(\alpha, \beta, \gamma) = \frac{\big(e^\alpha, \, e^\beta, \, e^\gamma\big)}{e^\alpha + e^\beta + e^\gamma}. \tag{5}$$

so that $\tilde{\alpha} + \tilde{\beta} + \tilde{\gamma} = 1$. This formulation allows the model to automatically adjust the importance of each geometry during training while avoiding trivial solutions and ensuring stability.

By the joint learning of curvature $(c_H, c_S)$ and weights $(\tilde{\alpha}, \tilde{\beta}, \tilde{\gamma})$, the product space dynamically adapts to the intrinsic geometry of the data. This flexibility enables the synthetic data distribution to be aligned with the real data distribution under the most suitable geometric representation.

Besides, it is worth mentioning that our framework is agnostic to the specific choice of the distribution matching method. Although we adopt the NCFM method for concreteness, other distribution matching approaches can also be seamlessly integrated. As shown in Figure 3, replacing NCFM with alternative matching schemes does not affect the improvement, demonstrating the robustness of our framework to the choice of the distribution matching objective.

## 3.2 Optimal Transport

To improve the representational faithfulness of the distilled data set, we design an OT loss to measure the correspondence between real and synthetic samples across the Euclidean, hyperbolic, and spherical components of the product manifold (Villani et al., 2008). This loss is used as an auxiliary objective to maintain the class-wise mass and avoid collapse, i.e., a single geometry dominates.

Let $\mathcal{X}$ denote the original data manifold and $\mathcal{X}_{\text{syn}}$ the synthetic data manifold, with corresponding probability measures $\rho \in \mathcal{P}(\mathcal{X})$ and $\rho_{\text{syn}} \in \mathcal{P}(\mathcal{X}_{\text{syn}})$. We construct an embedding

$$f_\theta : \mathcal{X} \longrightarrow \mathcal{P} = \mathbb{E}^{d_E} \times \mathbb{H}_{c_H}^{d_H} \times \mathbb{S}_{c_S}^{d_S},$$

For embedded points $z = (z^E, z^H, z^S)$ and $z' = (\tilde{z}^E, \tilde{z}^H, \tilde{z}^S)$ in $\mathcal{P}$, we use Eq. (4) as our product-space metric. When optimizing the OT loss with respect to the synthetic embeddings $z$, we compute the Riemannian gradient $\text{grad}_z \, \mathcal{L}_{\text{OT}}$ in the tangent space $T_z \mathcal{P}$, obtained by projecting the Euclidean derivative onto $T_z \mathcal{P}$ under the product metric $g^{\mathcal{P}}$. A descent step is taken in $T_z \mathcal{P}$, and the result is mapped back to the manifold via the exponential map, so the updated embedding remains inside $\mathcal{P}$. The OT loss is then defined as the squared 2-Wasserstein distance between the push-forward measures $\mu = f_\theta \# \rho$ and $\nu = f_\theta \# \rho_{\text{syn}}$:

$$\mathcal{L}_{\text{OT}}(\rho, \rho_{\text{syn}}) = W_2^2(\mu, \nu) = \inf_{\pi \in \Pi(\mu, \nu)} \int_{\mathcal{P} \times \mathcal{P}} d_{\mathcal{P}}^2(z, \tilde{z}) \, d\pi(z, \tilde{z}), \tag{6}$$

where $\Pi(\mu, \nu)$ is the set of couplings with marginals $\mu$ and $\nu$, $d\pi(z, \tilde{z})$ indicates integration with respect to the coupling $\pi$. During training, this OT loss is combined with the primary distribution-matching objective, and gradients for the hyperbolic and spherical components are computed in their tangent representations and mapped back through exponential maps, ensuring that the learned embeddings of synthetic data remain valid manifold points throughout optimization.

## 3.3 OVERALL OBJECTIVE

To guide the optimization of the distilled dataset, we integrate all objectives into a single training criterion that balances distribution matching, geometry-aware transport, and curvature regularization. Specifically, the overall loss is defined as

$$\mathcal{L}_{\text{total}} = \mathcal{L}_{\text{DM}} + \lambda_{\text{OT}}\mathcal{L}_{\text{OT}} + \lambda_{\text{curv}}\mathcal{L}_{\text{curv}}, \tag{7}$$

where $\mathcal{L}_{\text{DM}}$ aligns the real and synthetic data distributions in the product space, $\mathcal{L}_{\text{OT}}$ further refines this alignment via OT computed in the tangent spaces of curved components, and $\mathcal{L}_{\text{curv}}$ stabilizes the learnable curvature parameters $\{c_H, c_S\}$. In addition, learnable weights $(\tilde{\alpha}, \tilde{\beta}, \tilde{\gamma})$ regulate the relative contributions of the Euclidean, hyperbolic, and spherical components. By minimizing $\mathcal{L}_{\text{total}}$ via gradient-based optimization methods, the synthetic data progressively approximate the real data distribution while retaining the intrinsic manifold geometry, with the expectation of strong generalization across diverse tasks.

## 4 THEORETICAL ANALYSIS

Given the proposed framework above, we are wondering if our intuition could be theoretically justified on why dataset distillation benefits from incorporating non-Euclidean geometry and why modeling data in a product space helps improve performance.

**Notions and notations.** We denote by $(\mathcal{M}, d_{\mathcal{M}})$ the intrinsic data manifold with geodesic metric $d_{\mathcal{M}}$, and by $(\mathcal{X}_\kappa, d_{\mathcal{X}_\kappa})$ the matching space (Euclidean, hyperbolic, or spherical) with its canonical metric. A representation map $\Phi_\kappa : \mathcal{M} \to \mathcal{X}_\kappa$ induces the push-forward measure $\Phi_{\kappa\#}\rho(B) = \rho(\Phi_\kappa^{-1}(B))$ for any distribution $\rho$ on $\mathcal{M}$. The population risk is $R(\theta; \mu) = \mathbb{E}_{(z,y)\sim\mu}[\ell(f_\theta(z), y)]$, and $\text{IPM}^{(\mathcal{X}_\kappa)}$ denotes an integral probability metric on $(\mathcal{X}_\kappa, d_{\mathcal{X}_\kappa})$. We write $\mu_y$ for the class prior of label $y$. Finally, the geometric distortion introduced by $\Phi_\kappa$ is controlled by a multiplicative factor $\beta(\kappa)$ and an additive residual $\varepsilon_{\text{dist}}(\kappa)$, which measures how faithfully the matching space preserves the curvature of the original manifold.

**Assumption 4.1** (Regularity and convergence)**.**

i) *(Data manifold)* The real data distribution $\mu$ is supported on a measurable manifold $(\mathcal{M}, d_{\mathcal{M}})$.

ii) *(Lipschitz loss)* For all $\theta$ and $y$, the loss is $L$-Lipschitz with respect to the input:

$$|\ell(f_\theta(z), y) - \ell(f_\theta(z'), y)| \leq L\, d_{\mathcal{X}_\kappa}(z, z'), \quad \forall z, z' \in \mathcal{X}_\kappa.$$

iii) *(Algorithmic stability)* The training algorithm $\mathcal{A}$ is uniformly stable with constant $\varepsilon_{\text{stab}}$, i.e.,

$$\left| R(\theta^{\rho_1}; \mu) - R(\theta^{\rho_2}; \mu) \right| \leq \varepsilon_{\text{stab}}, \quad \forall \rho_1, \rho_2.$$

iv) *(Statistical convergence)* The empirical distilled distribution $\nu_M$ satisfies, for every label $y$,

$$\text{IPM}^{(\mathcal{X}_\kappa)}\big(\Phi_{\kappa\#}\mu(\cdot|y),\ \Phi_{\kappa\#}\nu_M(\cdot|y)\big) \leq \mathcal{E}_{\text{stat}}^{(y)}(M) + \varepsilon_{\text{opt}},$$

where $\mathcal{E}_{\text{stat}}^{(y)}(M) \to 0$ as $M \to \infty$. Equivalently, the class-weighted average satisfies

$$\sum_y \mu_y\, \text{IPM}^{(\mathcal{X}_\kappa)}\big(\Phi_{\kappa\#}\mu(\cdot|y),\ \Phi_{\kappa\#}\nu_M(\cdot|y)\big) \leq \mathcal{E}_{\text{stat}}(M) + \varepsilon_{\text{opt}},$$

with $\mathcal{E}_{\text{stat}}(M) \to 0$ as $M \to \infty$. Here "#" denotes the push-forward measure, i.e., $\Phi_{\kappa\#}\mu(B) = \mu(\Phi_\kappa^{-1}(B))$ for all measurable $B \subseteq \mathcal{X}_\kappa$.

**Theorem 4.1** (Geometry-driven risk decomposition). *Let $\mu$ be the real data distribution and $\nu_M$ the distilled data distribution of size $M$. Then*

$$\Delta \;\leq\; L\Big(\beta(\kappa) \cdot \big[\overline{\mathcal{E}}_{\text{stat}}(M) + \varepsilon_{\text{opt}}\big] + C\,\varepsilon_{\text{dist}}(\kappa)\Big) \;+\; \varepsilon_{\text{stab}},$$

*where $\overline{\mathcal{E}}_{\text{stat}}(M) := \sum_y \mu_y\,\mathcal{E}_{\text{stat}}^{(y)}(M)$ is the class-weighted average statistical error, and $C$ is a universal constant depending only on the chosen IPM.*

Theorem 4.1 establishes that the generalization gap of dataset distillation can be decomposed into statistical $\overline{\mathcal{E}}_{\text{stat}}$, stability $\varepsilon_{\text{stab}}$, and geometric $\varepsilon_{\text{dist}}$ terms. Particularly, the geometry plays a critical role: if the matching space fails to capture the curvature of the underlying data manifold, the bound can degrade significantly. To improve the bound in this case, we assume that the real data distribution $\mu$ is supported on a mixed-curvature product manifold

$$\mathcal{M}^{\star} \;\subset\; \mathbb{E}^{d_E} \times \mathbb{H}_{c_H}^{d_H} \times \mathbb{S}_{c_S}^{d_S},$$

contained in a geodesic ball of radius $R$. This theorem is both mathematically natural and empirically grounded: hyperbolic spaces are known to embed taxonomies with exponentially smaller distortion than Euclidean spaces (Sarkar, 2011; Verbeek & Suri, 2014), spherical spaces are the canonical setting for directional statistics and cyclic representations (Mardia & Jupp, 2009), and Euclidean spaces provide accurate local approximations by standard Riemannian arguments. Therefore, we can assume that real-world datasets rarely conform to a purely Euclidean geometry but instead exhibit a mixture of flat, hierarchical, and angular structures.

Under the assumption, we show that product spaces of constant curvature yield provably tighter guarantees than Euclidean spaces alone. Specifically:

**Theorem 4.2** (Product spaces yield tighter bounds). *Under Assumption 4.1, there exists a constant $\delta > 0$ such that the generalization gap of distillation in the product space satisfies*

$$\Delta_{\text{product}} \;\leq\; \Delta_{\text{Euclid}} - L\,\delta,$$

*where $\Delta_{\text{Euclid}}$ denotes the Euclidean-space bound in Theorem 4.1.*

The improvement $\delta$ comes from two sources: for hierarchical data, Euclidean embeddings suffer exponentially growing distortion with radius $\tau$, whereas hyperbolic embeddings represent such structure isometrically; for angular or periodic data, Euclidean embeddings reduce geodesic arcs to chords, introducing a fixed distortion gap (at least a $\pi/2$ factor), while spherical embeddings preserve the geometry exactly. Flat data incurs no loss in either case. By combining these advantages, product manifolds reduce both multiplicative distortion $\mathcal{E}_{\text{stat}}$ or $\varepsilon_{\text{stab}}$, and additive residual terms $\varepsilon_{\text{dist}}$, giving rise to strictly tighter guarantees than Euclidean-only matching. A complete proof, including hyperbolic packing arguments and spherical arc–chord inequalities, is provided in Appendix A.

# 5 EXPERIMENTS

## 5.1 EXPERIMENTAL SETUP

**Datasets and Baselines.** To validate the efficiency of our work, we conduct experiments on three datasets, including MNIST (LeCun et al., 1998), CIFAR-10 (Krizhevsky et al., 2009), and CIFAR-100 (Krizhevsky et al., 2009). We compare our method, GeoDM, against a broad range of baselines. Gradient-matching approaches include DC (Zhao et al., 2020b), DCC (Lee et al., 2022), and DSA (Zhao & Bilen, 2021). Distribution-matching methods considered are CAFE (Wang et al., 2022), DM (Zhao & Bilen, 2023), IDM (Zhao et al., 2023), M3D (Zhang et al., 2024), IID (Deng et al., 2024), DREAM (Liu et al., 2023b), DSDM (Li et al., 2024), G-VBSM (Shao et al., 2024), and NCFM (Wang et al., 2025). We also evaluate against trajectory-matching techniques such as MTT (Cazenavette et al., 2022), FTD (Du et al., 2023), and ATT (Liu et al., 2024). In addition, we include coreset-selection baselines, namely random selection, Herding (Welling, 2009; Rebuffi et al., 2017), and K-center (Farahani & Hekmatfar, 2009; Sener & Savarese, 2017).

Table 1: Comparison of dataset distillation methods on MNIST, CIFAR-10, and CIFAR-100. "Whole" denotes the test accuracy of a model trained on the full dataset. Note that methods marked with $^\dagger$ are quoted from Wang et al. (2025)

| Datasets | MNIST | | | CIFAR-10 | | | CIFAR-100 | | |
|---|---|---|---|---|---|---|---|---|---|
| IPC | 1 | 10 | 50 | 1 | 10 | 50 | 1 | 10 | 50 |
| Ratio | 0.017% | 0.17% | 0.83% | 0.02% | 0.2% | 1% | 0.2% | 2% | 10% |
| Whole | | 99.6 ± 0.0 | | | 84.8 ± 0.1 | | | 56.2 ± 0.3 | |
| Random $^\dagger$ | 64.9 ± 3.5 | 95.1 ± 0.9 | 97.9 ± 0.5 | 14.4 ± 2.0 | 26.0 ± 1.2 | 43.4 ± 1.0 | 1.4 ± 0.1 | 5.0 ± 0.2 | 15.0 ± 0.4 |
| Herding $^\dagger$ | 89.2 ± 1.6 | 93.7 ± 0.3 | 94.8 ± 0.3 | 21.5 ± 1.2 | 31.6 ± 0.7 | 40.4 ± 0.6 | 8.4 ± 0.3 | 17.3 ± 0.3 | 33.7 ± 0.5 |
| K-center | 89.3 ± 1.5 | 84.4 ± 1.7 | 97.4 ± 0.3 | 21.5 ± 1.3 | 14.7 ± 0.7 | 27.0 ± 1.4 | 8.3 ± 0.3 | 7.1 ± 0.2 | 30.5 ± 0.3 |
| DC | 91.7 ± 0.5 | 97.4 ± 0.3 | 98.8 ± 0.2 | 28.3 ± 0.5 | 44.9 ± 0.5 | 53.9 ± 0.5 | 12.8 ± 0.3 | 25.2 ± 0.3 | 31.5 ± 0.2 |
| DCC | - | - | - | 32.9 ± 0.8 | 49,4 ± 0.5 | 61.6 ± 0.4 | 13.3 ± 0.3 | 30.6 ± 0.4 | 40.0 ± 0.3 |
| DSA | 88.7 ± 0.6 | 97.8 ± 0.1 | 99.2 ± 0.1 | 28.8 ± 0.7 | 52.1 ± 0.5 | 60.6 ± 0.5 | 13.9 ± 0.3 | 32.3 ± 0.3 | 42.8 ± 0.4 |
| MTT | - | - | - | 46.3 ± 0.8 | 65.3 ± 0.7 | 71.6 ± 0.7 | 24.3 ± 0.3 | 40.1 ± 0.4 | 47.7 ± 0.2 |
| ATT | - | - | - | 48.3 ± 0.8 | 67.7 ± 0.7 | 74.5 ± 0.7 | 26.1 ± 0.3 | 44.2 ± 0.4 | 51.2 ± 0.2 |
| FTD | - | - | - | 46.8 ± 0.8 | 66.6 ± 0.7 | 73.8 ± 0.7 | 25.2 ± 0.3 | 43.4 ± 0.4 | 50.7 ± 0.2 |
| DM | 89.7 ± 0.6 | 97.5 ± 0.1 | 98.6 ± 0.1 | 30.3 ± 0.1 | 48.9 ± 0.6 | 63.0 ± 0.4 | 11.4 ± 0.3 | 29.7 ± 0.3 | 43.6 ± 0.4 |
| CAFE | 93.1 ± 0.3 | 97.2 ± 0.2 | 98.6 ± 0.2 | 42.6 ± 3.3 | 46.3 ± 0.6 | 55.5 ± 0.6 | 12.9 ± 0.3 | 27.8 ± 0.3 | 37.9 ± 0.3 |
| IDM | - | - | - | 45.6 ± 0.7 | 58.6 ± 0.1 | 67.5 ± 0.1 | 20.1 ± 0.3 | 45.1 ± 0.1 | 50.0 ± 0.2 |
| M3D | 94.4 ± 0.2 | 97.6 ± 0.1 | 98.2 ± 0.3 | 45.3 ± 0.5 | 63.5 ± 0.2 | 69.9 ± 0.5 | 26.2 ± 0.3 | 42.4 ± 0.2 | 50.9 ± 0.7 |
| IID | - | - | - | 47.1 ± 0.1 | 59.9 ± 0.1 | 69.0 ± 0.3 | 24.6 ± 0.1 | 45.7 ± 0.4 | 51.3 ± 0.4 |
| DREAM | 95.3 ± 0.2 | 98.4 ± 0.2 | 99.0 ± 0.1 | 50.4 ± 0.3 | 67.2 ± 0.2 | 73.3 ± 0.6 | 26.7 ± 0.2 | 43.0 ± 0.4 | 48.5 ± 0.5 |
| DSDM | 94.5 ± 0.4 | 98.1 ± 0.3 | 98.7 ± 0.2 | 44.3 ± 0.4 | 66.1 ± 0.3 | 75.3 ± 0.3 | 19.2 ± 0.4 | 46.0 ± 0.2 | 53.1 ± 0.5 |
| G-VBSM $^\dagger$ | - | - | - | - | 46.5 ± 0.7 | 54.3 ± 0.3 | 16.4 ± 0.7 | 38.7 ± 0.2 | 45.7 ± 0.4 |
| NCFM | 93.4 ± 0.5 | 95.7 ± 0.6 | 96.9 ± 0.3 | 49.5 ± 0.3 | 71.8 ± 0.5 | 77.4 ± 0.3 | 34.4 ± 0.5 | 48.4 ± 0.3 | 54.7 ± 0.2 |
| **GeoDM** | **96.3 ± 0.2** | **98.6 ± 0.2** | **99.3 ± 0.1** | **51.2 ± 0.2** | **74.4 ± 0.3** | **78.3 ± 0.2** | **38.0 ± 0.4** | **49.2 ± 0.3** | **55.0 ± 0.2** |

**Configuration.** We follow prior work (Wang et al., 2025) to conduct our experiments. Unless otherwise specified, we use a 3-layer convolutional network as the backbone. The matching optimization is performed for $T = 10^4$ iterations. The weight for the OT loss $\lambda_{OT}$ is set to 2, while the weight for the curvature regularization $\lambda_{curv}$ is set to 1. Both the hyperbolic and spherical curvatures are initialized uniformly at random, along with the weights of each factor space. For evaluation, models are trained for 1500 epochs. We use accuracy as the evaluation metric across all experiments. All experiments are repeated three times, with mean and standard deviation of results (refer to Appendix E for results under more settings).

## 5.2 RESULT ANALYSIS

Table 1 summarizes the test accuracy (%) of representative dataset distillation methods on MNIST, CIFAR-10, and CIFAR-100 under different images-per-class (IPC) budgets. On MNIST, **GeoDM** achieves 96.3% accuracy at IPC = 1, improving over the best baseline (M3D, 94.4%) by about 2%. On CIFAR-10, GeoDM attains 74.4% at IPC = 10, surpassing state-of-the-art by approximately 3%. For the more challenging CIFAR-100, GeoDM reaches 38.0% and 49.2% at IPC = 1 and 10, yielding gains of about 3% and 1% respectively. These results demonstrate more than incremental improvements: the consistent gains at low IPC indicate that incorporating curvature information allows the distilled synthetic images to encode richer geometric structure, compensating for severe data scarcity. As IPC increases, however, the relative improvement gradually diminishes, since the abundance of synthetic samples reduces the need for additional structural information.

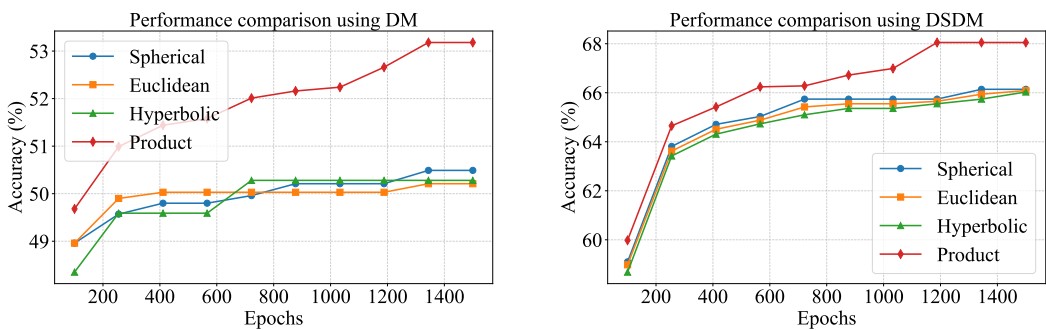

Figure 3: Comparison between product spaces and single spaces on CIFAR-10 with IPC=10.

Table 2: Cross-architecture resuls on CIFAR10.

| IPC | Evaluation | DREAM | M3D | DSDM | NCFM | Ours |
|-----|-----------|-------|-----|------|------|------|
| 10 | ConvNet-3 | 67.2±0.2 | 63.5±0.2 | 66.1±0.3 | 70.4±0.5 | **74.4±0.3** |
| | ResNet-18 | 64.7±0.2 | 29.9±0.3 | 51.7±0.2 | 67.3±0.4 | **69.1±0.5** |
| | AlexNet | 65.1±0.3 | 34.5±0.1 | 47.7±0.5 | 67.4±0.5 | **69.0±0.3** |
| 50 | ConvNet-3 | 73.3±0.6 | 69.9±0.5 | 75.3±0.3 | 77.4±0.3 | **78.3±0.2** |
| | ResNet-18 | 69.3±0.4 | 31.6±0.9 | 61.4±0.3 | 73.7±0.2 | **74.5±0.4** |
| | AlexNet | 72.3±0.1 | 35.6±0.8 | 48.2±0.2 | 75.5±0.3 | **75.6±0.1** |

**Robustness.** Figure 3 shows that our experimental results further validate the effectiveness of product-space modeling. Specifically, when replacing the distribution matching component with a simple DM objective (Zhao & Bilen, 2023), our method achieves up to 2.79% improvement over single-space baselines. Even when adopting a stronger DSDM formulation (Li et al., 2024), the performance gain remains around 2%. These results highlight that the product space is capable of capturing richer geometric information in the features, enabling synthetic images to retain curvature properties that would otherwise be neglected. Beyond absolute accuracy gains, Moreover, the consistent advantages across different DM variants confirm the generality of our framework and echo our theoretical analysis, indicating that the benefit of product-space modeling is intrinsic to the alignment process rather than tied to a specific instantiation of distribution matching.

**Cross-architecture** Table 2 reports the cross-architecture evaluation results on CIFAR-10 with IPC values of 10 and 50. In this setting, we first distilled synthetic datasets using ConvNet-3 as the condensation model, and then trained two different architectures, ResNet-18 and AlexNet, on the distilled data. The performance was evaluated on the CIFAR-10 test set. Across both IPC settings, our method consistently achieves the best results when transferring to deeper or alternative architectures, demonstrating stronger generalization ability of the distilled data.

**Ablation study.** Table 3 presents the ablation study on CIFAR-10 with IPC = 10. We first note that the product space alone already delivers a clear improvement (73.5% vs. 71.8% for single-geometry modeling), confirming that combining multiple geometries inherently captures richer structural information than a flat Euclidean embedding. Adding curvature adaptation and learnable weights further improves performance by making the synthetic embeddings better follow the underlying data manifold, rather than being confined to fixed-curvature assumptions. Finally, incorporating the OT loss provides consistent transport between real and synthetic distributions while preventing dominance of any single geometry, resulting in the best performance of 74.4% when all components are combined.

Table 3: Ablation study

| Product | Curv & Weight | OT | CIFAR-10 |
|---------|---------------|-----|----------|
| | | | 71.8±0.3 |
| ✓ | | | 73.5±0.2 |
| | | ✓ | 72.3±0.1 |
| ✓ | ✓ | | 73.9±0.2 |
| ✓ | | ✓ | 73.8±0.1 |
| ✓ | ✓ | ✓ | **74.4±0.2** |

## 6 CONCLUSION

In this work, we revisit dataset distillation from the perspective of manifold learning. Existing distribution-matching approaches largely operate in Euclidean feature spaces, which fail to capture the intrinsic geometry of real-world data. To address this, we propose GeoDM, a geometry-aware framework that performs matching in a Cartesian product of Euclidean, hyperbolic, and spherical manifolds. By modeling varying curvature, introducing learnable weights across geometries, and designing an optimal transport loss, our method achieves more faithful manifold-aware alignment. Moreover, our theoretical analysis shows that product-space matching provides tighter error bounds than Euclidean baselines. Extensive experiments on benchmark datasets further validate our approach and support the theoretical findings. A limitation of our work is that the dimensionality of each manifold is fixed; future research could explore incorporating learnable dimensions to better fuse multiple geometries without sacrificing performance.

## ETHICS STATEMENT

This work adheres to the ICLR Code of Ethics and complies with the principles of responsible research conduct. Our study does not involve human subjects, personal data, or other sensitive information requiring ethics approval. All datasets are publicly available and appropriately licensed, with proper citation to their original sources. We provide our implementation and experimental settings to promote transparency, reproducibility, and community validation. Finally, no conflicts of interest or external sponsorships have influenced the design, execution, or reporting of this study.

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

# A PROOF OF THEORY SECTION

## A.1 PROOF OF GEOMETRY-DRIVEN RISK DECOMPOSITION THEOREM 4.1

**Theorem A.1** (Geometry-driven risk decomposition repeat). *Let $\mu$ be the real distribution and $\nu_M$ the distilled distribution of size $M$. The excess risk satisfies*

$$\Delta \triangleq \big| R(\theta^\nu; \mu) - R(\theta^\mu; \mu) \big| \leq L\Big( \beta(\kappa) \cdot \big[ \overline{\mathcal{E}}_{\text{stat}}(M) + \varepsilon_{\text{opt}} \big] + C\, \varepsilon_{\text{dist}}(\kappa) \Big) + \varepsilon_{\text{stab}},$$

*where $\overline{\mathcal{E}}_{\text{stat}}(M) := \sum_y \mu_y\, \mathcal{E}_{\text{stat}}^{(y)}(M)$ is the class-weighted average statistical error, and $C$ is a universal constant depending only on the chosen IPM. (for $W_1$, one can take $C = 1$).*

*Proof.* **Step 0 (Notation and setup).** Recall the population risk $R(\theta; \rho) = \mathbb{E}_{(x,y)\sim\rho}[\ell(f_\theta(x), y)]$, and let $\Delta := |R(\theta^\nu; \mu) - R(\theta^\mu; \mu)|$. Write $\Phi_\kappa : \mathcal{M} \to \mathcal{X}_\kappa$ for the representation map, and denote push-forward measures on $\mathcal{X}_\kappa$ by $\Phi_{\kappa\#}\rho$: $\Phi_{\kappa\#}\rho(B) = \rho(\Phi_\kappa^{-1}(B))$ for measurable $B \subseteq \mathcal{X}_\kappa$.

**Step 1 (Triangle inequality).** Add and subtract $R(\theta^\nu; \nu_M)$ and use the triangle inequality:

$$\begin{aligned}
\Delta &= \big| R(\theta^\nu; \mu) - R(\theta^\mu; \mu) \big| \\
&\leq \underbrace{\big| R(\theta^\nu; \mu) - R(\theta^\nu; \nu_M) \big|}_{\text{(A) distribution shift under fixed } \theta^\nu} + \underbrace{\big| R(\theta^\nu; \nu_M) - R(\theta^\mu; \mu) \big|}_{\text{(B) algorithmic stability}}.
\end{aligned} \tag{8}$$

**Step 2 (Bounding term (A) via Lipschitz loss and $W_1$ on $\mathcal{X}_\kappa$).** Fix $\theta = \theta^\nu$ and define $h(z, y) := \ell(f_\theta(z), y)$ for $z \in \mathcal{X}_\kappa$. By Assumption 4.1 (ii), for each $y$ the map $z \mapsto h(z, y)$ is $L$-Lipschitz:

$$|h(z, y) - h(z', y)| \leq L\, d_{\mathcal{X}_\kappa}(z, z'), \quad \forall z, z' \in \mathcal{X}_\kappa.$$

We assume label-prior matching, i.e., $\mu_y = \nu_y$, so that the label marginals coincide and only the conditionals $\mu(\cdot|y)$ and $\nu_M(\cdot|y)$ may differ. Then

$$\begin{aligned}
\big| R(\theta^\nu; \mu) - R(\theta^\nu; \nu_M) \big| &= \Big| \mathbb{E}_{y\sim\mu_y}\big[ \mathbb{E}_{x\sim\mu(\cdot|y)} h(\Phi_\kappa(x), y) - \mathbb{E}_{x\sim\nu_M(\cdot|y)} h(\Phi_\kappa(x), y) \big] \Big| \\
&\leq \mathbb{E}_{y\sim\mu_y} \Big| \mathbb{E}_{x\sim\mu(\cdot|y)} h(\Phi_\kappa(x), y) - \mathbb{E}_{x\sim\nu_M(\cdot|y)} h(\Phi_\kappa(x), y) \Big|.
\end{aligned}$$

For each fixed $y$, by the Kantorovich–Rubinstein duality on $(\mathcal{X}_\kappa, d_{\mathcal{X}_\kappa})$ (Villani et al., 2008),

$$\Big| \mathbb{E}_{\mu(\cdot|y)} h(\Phi_\kappa(x), y) - \mathbb{E}_{\nu_M(\cdot|y)} h(\Phi_\kappa(x), y) \Big| \leq L\, W_1^{(\mathcal{X}_\kappa)}\big( \Phi_{\kappa\#}\mu(\cdot|y), \Phi_{\kappa\#}\nu_M(\cdot|y) \big).$$

Averaging over $y$ with weights $\mu_y = \nu_y$ yields the *class-wise* bound

$$\big| R(\theta^\nu; \mu) - R(\theta^\nu; \nu_M) \big| \leq L \sum_y \mu_y\, W_1^{(\mathcal{X}_\kappa)}\big( \Phi_{\kappa\#}\mu(\cdot|y), \Phi_{\kappa\#}\nu_M(\cdot|y) \big). \tag{9}$$

We denote the right-hand side by

$$\overline{W}_{1,y}^{(\mathcal{X}_\kappa)}(\mu, \nu_M) := \sum_y \mu_y\, W_1^{(\mathcal{X}_\kappa)}\big( \Phi_{\kappa\#}\mu(\cdot|y), \Phi_{\kappa\#}\nu_M(\cdot|y) \big),$$

so that Eq. (9) is equivalently $\big| R(\theta^\nu; \mu) - R(\theta^\nu; \nu_M) \big| \leq L\, \overline{W}_{1,y}^{(\mathcal{X}_\kappa)}(\mu, \nu_M)$.

**Step 3 (Relating class-wise $W_1$ on $\mathcal{X}_\kappa$ to $\mathcal{M}$ via geometric distortion).** For each label $y$, use the *primal* definition of $W_1$:

$$W_1^{(\mathcal{X}_\kappa)}\big(\Phi_{\kappa\#}\mu(\cdot|y), \Phi_{\kappa\#}\nu_M(\cdot|y)\big) = \inf_{\pi \in \Pi(\mu(\cdot|y),\nu_M(\cdot|y))} \int d_{\mathcal{X}_\kappa}\big(\Phi_\kappa(x), \Phi_\kappa(x')\big) \, d\pi(x, x').$$

By the geometric distortion inequality Eq. (9) (Tenenbaum et al., 2000), for any coupling $\pi$,

$$d_{\mathcal{X}_\kappa}\big(\Phi_\kappa(x), \Phi_\kappa(x')\big) \leq \beta(\kappa) \, d_{\mathcal{M}}(x, x') + \varepsilon_{\text{dist}}(\kappa).$$

Integrating both sides w.r.t. $\pi$ and taking the infimum yields

$$W_1^{(\mathcal{X}_\kappa)}\big(\Phi_{\kappa\#}\mu(\cdot|y), \Phi_{\kappa\#}\nu_M(\cdot|y)\big) \leq \beta(\kappa) \, W_1^{(\mathcal{M})}\big(\mu(\cdot|y), \nu_M(\cdot|y)\big) + \varepsilon_{\text{dist}}(\kappa).$$

Weighting by $\mu_y$ and summing over $y$ gives

$$\big|R(\theta^\nu; \mu) - R(\theta^\nu; \nu_M)\big| \leq L\Big(\beta(\kappa) \, \overline{W}_{1,y}^{(\mathcal{M})}(\mu, \nu_M) + \varepsilon_{\text{dist}}(\kappa)\Big), \tag{10}$$

where

$$\overline{W}_{1,y}^{(\mathcal{M})}(\mu, \nu_M) := \sum_y \mu_y \, W_1^{(\mathcal{M})}\big(\mu(\cdot|y), \nu_M(\cdot|y)\big).$$

**Step 4 (Bounding term (B) via uniform stability).** By Assumption 4.1 (iii), the training algorithm $\mathcal{A}$ is uniformly stable with constant $\varepsilon_{\text{stab}}$ in the sense of Bousquet & Elisseeff (2002):

$$\big|R(\theta^{\rho_1}; \rho) - R(\theta^{\rho_2}; \rho)\big| \leq \varepsilon_{\text{stab}}, \quad \forall \rho_1, \rho_2, \rho.$$

Setting $\rho_1 = \nu_M$, $\rho_2 = \mu$, and evaluating on $\rho = \mu$ gives

$$\big|R(\theta^\nu; \nu_M) - R(\theta^\mu; \mu)\big| \leq \varepsilon_{\text{stab}}. \tag{11}$$

**Step 5 (Statistical convergence of the class-wise IPM on $\mathcal{M}$).** By Assumption 4.1 (iv), for each $y$ there exists a decreasing function $\mathcal{E}_{\text{stat}}^{(y)}(M)$ and an optimization/modeling error $\varepsilon_{\text{opt}}$ such that

$$W_1^{(\mathcal{M})}\big(\mu(\cdot|y), \nu_M(\cdot|y)\big) \leq \mathcal{E}_{\text{stat}}^{(y)}(M) + \varepsilon_{\text{opt}}.$$

Hence the class-weighted average satisfies

$$\overline{W}_{1,y}^{(\mathcal{M})}(\mu, \nu_M) \leq \overline{\mathcal{E}}_{\text{stat}}(M) + \varepsilon_{\text{opt}}, \qquad \overline{\mathcal{E}}_{\text{stat}}(M) := \sum_y \mu_y \mathcal{E}_{\text{stat}}^{(y)}(M). \tag{12}$$

**Step 6 (Combine all bounds).** Combining Eq. (10), Eq. (11), and Eq. (12) yields

$$\Delta \leq L\Big(\beta(\kappa) \cdot \big[\overline{\mathcal{E}}_{\text{stat}}(M) + \varepsilon_{\text{opt}}\big] + \varepsilon_{\text{dist}}(\kappa)\Big) + \varepsilon_{\text{stab}}.$$

This matches the claimed inequality with $C = 1$ for $W_1$. For other IPMs (e.g., MMD), the same derivation holds with a geometry-dependent constant $C$ induced by the kernel's Lipschitz/smoothness envelope on $(\mathcal{X}_\kappa, d_{\mathcal{X}_\kappa})$ (Sriperumbudur et al., 2010).

$\square$

## A.2 Detailed Proofs for Product-Space Advantage

**Product geometry and metrics.** We equip product spaces with the $\ell_2$ product metric unless stated otherwise. If $\mathcal{X} = \mathcal{X}_1 \times \cdots \times \mathcal{X}_K$ and $d_{\mathcal{X}}^2 = \sum_{k=1}^K d_{\mathcal{X}_k}^2$, then for a representation $\Phi = (\Phi^{(1)}, \ldots, \Phi^{(K)})$ with component-wise distortions

$$d_{\mathcal{X}_k}\big(\Phi^{(k)}(x), \Phi^{(k)}(x')\big) \leq \beta_k \, d_{\mathcal{M}}(x, x') + \varepsilon_k,$$

we have (by Minkowski/triangle inequality) (Yosida, 2012)

$$d_{\mathcal{X}}\big(\Phi(x), \Phi(x')\big) \leq \sqrt{\sum_{k=1}^K \big(\beta_k \, d_{\mathcal{M}}(x, x') + \varepsilon_k\big)^2} \leq \Big(\max_k \beta_k\Big) d_{\mathcal{M}}(x, x') + \sum_{k=1}^K \varepsilon_k, \tag{13}$$

so that we may take $\beta_{\text{prod}} = \max_k \beta_k$ and $\varepsilon_{\text{dist,prod}} = \sum_k \varepsilon_k$.

**Two canonical $W_1$ representations.** We will repeatedly use (i) the primal $W_1$ formulation via couplings:

$$W_1(\alpha, \beta) = \inf_{\pi \in \Pi(\alpha, \beta)} \int d(u, v) \, d\pi(u, v),$$

and (ii) the dual KR characterization $W_1(\alpha, \beta) = \sup_{\|g\|_{\text{Lip}} \leq 1} \big|\mathbb{E}_\alpha g - \mathbb{E}_\beta g\big|$ (Villani et al., 2008)

### A.2.1 HYPERBOLIC FACTOR: LOWER BOUNDS ON EUCLIDEAN DISTORTION

We first show that the hyperbolic component of the data manifold forces any Euclidean-only matching space to incur a multiplicative distortion that grows at least linearly in the geodesic radius $R$. The proof uses a standard chain: (a) large hyperbolic balls contain large tree-like subsets; (b) trees embed into hyperbolic space with $O(1)$ distortion; (c) finite trees require $\Omega(\log n)$ bi-Lipschitz distortion to embed into Euclidean/Hilbert spaces.

**Lemma A.2** (Hyperbolic volume growth and large separated subsets). *Let $B_{\mathbb{H}}(R)$ be a geodesic ball of radius $R$ in $\mathbb{H}^{d_H}_{c_H}$ with $c_H < 0$. Then there exist constants $C_1, \gamma > 0$ (depending only on $d_H$ and $|c_H|$) and a $\Delta > 0$ such that $B_{\mathbb{H}}(R)$ contains a $\Delta$ separated subset $\mathcal{P}_R$ with cardinality*

$$|\mathcal{P}_R| \;\geq\; C_1\, e^{\gamma R}.$$

*Proof.* **Step A.2.1 (Normalization and geodesic spheres).** By a standard rescaling, $\mathbb{H}^{d_H}_{c_H}$ is isometric to $\mathbb{H}^{d_H}_{-1}$ with distances multiplied by $\sqrt{|c_H|}^{-1}$, and $(d_H-1)$-dimensional measures multiplied by $|c_H|^{-(d_H-1)/2}$ (see, e.g., Ratcliffe (2006)). Therefore it suffices to prove the claim for curvature $-1$ and then rescale constants at the end. We henceforth work in $\mathbb{H}^d \equiv \mathbb{H}^{d_H}_{-1}$ and write $d$ for $d_H$.

Fix $R > 0$ and set $r := R/2$. Let $S(r)$ be the geodesic sphere of radius $r$ centered at some $o \in \mathbb{H}^d$, i.e., $S(r) = \{x \in \mathbb{H}^d : d_{\mathbb{H}}(o, x) = r\}$. Clearly $S(r) \subset B_{\mathbb{H}}(R)$.

**Step A.2.2 (Area of geodesic spheres).** The $(d-1)$-dimensional area of $S(r)$ equals

$$\mathrm{Area}(S(r)) \;=\; \omega_{d-1}\, \sinh^{d-1}(r),$$

where $\omega_{d-1}$ is the area of the unit sphere $\mathbb{S}^{d-1}$ (Ratcliffe, 2006). In particular, $\mathrm{Area}(S(r))$ grows like $c_d\, e^{(d-1)r}$ as $r \to \infty$ because $\sinh r = \frac{1}{2}(e^r - e^{-r}) \geq \frac{1}{2} e^{r-1}$ for $r \geq 1$.

**Step A.2.3 (From angular separation to hyperbolic separation).** Take two points $u, v \in S(r)$ with central angle $\theta \in [0, \pi]$ (i.e., the angle at $o$ between geodesics $ou$ and $ov$ is $\theta$). By the hyperbolic law of cosines (curvature $-1$) (Anderson, 2006),

$$\cosh d_{\mathbb{H}}(u,v) \;=\; \cosh^2 r \;-\; \sinh^2 r \cos\theta \;=\; 1 \;+\; \sinh^2 r\,(1 - \cos\theta).$$

Hence

$$\cosh d_{\mathbb{H}}(u,v) - 1 \;=\; \sinh^2 r\,(1 - \cos\theta).$$

Using the elementary inequality $\cosh t \geq 1 + \frac{t^2}{2}$ for all $t \in \mathbb{R}$, we obtain

$$\frac{d_{\mathbb{H}}(u,v)^2}{2} \;\leq\; \cosh d_{\mathbb{H}}(u,v) - 1 \;=\; \sinh^2 r\,(1 - \cos\theta).$$

Therefore
$$d_{\mathbb{H}}(u,v) \;\geq\; \sqrt{2}\, \sinh r\, \sqrt{1 - \cos\theta}. \tag{14}$$

Next, for $\theta \in [0, \pi]$, we have the elementary bound $1 - \cos\theta \;\geq\; \frac{2}{\pi^2}\theta^2$ (since $\cos\theta \leq 1 - 2\theta^2/\pi^2$ on $[0, \pi]$). Plugging this into Eq. (14) yields

$$d_{\mathbb{H}}(u,v) \;\geq\; \frac{2}{\pi}\, \sinh r\, \theta. \tag{15}$$

Consequently, if we enforce an *angular* separation $\theta \geq \theta_0$ with

$$\theta_0 \;:=\; \frac{\pi\, \Delta}{2\, \sinh r}, \tag{16}$$

then $d_{\mathbb{H}}(u,v) \geq \Delta$ for all such pairs on $S(r)$. Thus, constructing a $\theta_0$-separated set on $S(r)$ immediately gives a $\Delta$-separated set in $B_{\mathbb{H}}(R)$.

**Step A.2.4 (Maximal angular packing on $S(r)$).** Consider the intrinsic (Riemannian) metric on $S(r)$. A $\theta_0$-*separated set* $\mathcal{Q} \subset S(r)$ means that the central angle between any two distinct points of $\mathcal{Q}$ is at least $\theta_0$. Using a greedy maximality argument on the compact manifold $S(r)$, there exists a maximal $\theta_0$-separated set $\mathcal{Q}$ such that the collection of spherical caps $\{\,\mathrm{Cap}(q, \theta_0/2) : q \in \mathcal{Q}\,\}$ is pairwise disjoint, while the caps $\{\,\mathrm{Cap}(q, \theta_0) : q \in \mathcal{Q}\,\}$ cover $S(r)$ (Conway & Sloane, 2013, see

sphere packing arguments).Here, $\mathrm{Cap}(q, \varphi)$ denotes the subset of $S(r)$ with central angle at most $\varphi$ from $q$.

**Step A.2.5 (Area of spherical caps on $S(r)$).** The induced metric on $S(r)$ is (up to isometry) the round metric on a sphere of "radius" $\sinh r$, i.e., the area element scales by $\sinh^{d-1} r$ times that of $\mathbb{S}^{d-1}$. Consequently, the area of a cap of angular radius $\varphi \in (0, \pi)$ on $S(r)$ is

$$\mathrm{Area}\big(\mathrm{Cap}(\cdot, \varphi)\big) = \sinh^{d-1} r \cdot \omega_{d-2} \int_0^{\varphi} \sin^{d-2} t \, dt.$$

Using $\sin t \le t$ for $t \in [0, \pi]$, we obtain the *upper* bound

$$\mathrm{Area}\big(\mathrm{Cap}(\cdot, \varphi)\big) \le \sinh^{d-1} r \cdot \omega_{d-2} \int_0^{\varphi} t^{d-2} \, dt = \sinh^{d-1} r \cdot \frac{\omega_{d-2}}{d-1} \varphi^{d-1}. \quad (17)$$

**Step A.2.6 (Lower bound on the packing number on $S(r)$).** Let $N := |\mathcal{Q}|$ be the cardinality of a maximal $\theta_0$-separated set on $S(r)$. Because the caps $\{\mathrm{Cap}(q, \theta_0)\}_{q \in \mathcal{Q}}$ cover $S(r)$, we must have

$$\mathrm{Area}\big(S(r)\big) \le \sum_{q \in \mathcal{Q}} \mathrm{Area}\big(\mathrm{Cap}(q, \theta_0)\big) = N \cdot \mathrm{Area}\big(\mathrm{Cap}(\cdot, \theta_0)\big).$$

Therefore,

$$N \ge \frac{\mathrm{Area}\big(S(r)\big)}{\mathrm{Area}\big(\mathrm{Cap}(\cdot, \theta_0)\big)}. \quad (18)$$

Substituting $\mathrm{Area}(S(r)) = \omega_{d-1} \sinh^{d-1} r$ and using Eq. (17) with $\varphi = \theta_0$, we obtain

$$N \ge \frac{\omega_{d-1} \sinh^{d-1} r}{\sinh^{d-1} r \cdot \frac{\omega_{d-2}}{d-1} \theta_0^{d-1}} = \left(\frac{(d-1)\,\omega_{d-1}}{\omega_{d-2}}\right) \theta_0^{-(d-1)}. \quad (19)$$

Hence the packing number on $S(r)$ at angular resolution $\theta_0$ satisfies

$$N \ge C_d \, \theta_0^{-(d-1)}, \qquad C_d := \frac{(d-1)\,\omega_{d-1}}{\omega_{d-2}}.$$

**Step A.2.7 (From angular $\theta_0$ to metric $\Delta$ on $B_{\mathbb{H}}(R)$).** By Eq. (15) and the choice Eq. (16), any $\theta_0$-separated set $\mathcal{Q} \subset S(r)$ is $\Delta$-separated in the ambient hyperbolic metric. Therefore, with $\theta_0 = \frac{\pi \Delta}{2 \sinh r}$, inequality Eq. (19) gives

$$|\mathcal{P}_R| \ge N \ge C_d \left(\frac{2 \sinh r}{\pi \Delta}\right)^{d-1}.$$

**Step A.2.8 (Exponential growth in $R$).** For $r = R/2$ and $R \ge 2$, we have $\sinh r \ge \frac{1}{2} e^{r-1}$ and hence

$$|\mathcal{P}_R| \ge \tilde{C}_d \, \Delta^{-(d-1)} \, e^{(d-1)r} = \tilde{C}_d \, \Delta^{-(d-1)} \, e^{\frac{(d-1)}{2} R},$$

for some dimension-dependent constant $\tilde{C}_d > 0$. Thus there exist constants $C_1, \gamma > 0$ (depending only on $d$) and any fixed $\Delta \in (0, 1]$ such that

$$|\mathcal{P}_R| \ge C_1 \, e^{\gamma R}, \qquad \gamma = \frac{d-1}{2}.$$

**Step A.2.9 (Rescaling to curvature $c_H < 0$).** Returning to curvature $c_H < 0$, distances scale by $|c_H|^{-1/2}$ and areas by $|c_H|^{-(d-1)/2}$. Choosing the same (absolute) $\Delta > 0$ in the rescaled metric multiplies the angle threshold by a constant factor and therefore only alters $C_1$ and $\gamma$ by curvature-dependent constants. We may thus write the final bound with constants depending on $d_H$ and $|c_H|$:

$$|\mathcal{P}_R| \ge C_1(d_H, |c_H|) \, e^{\gamma(d_H, |c_H|) R}.$$

**Step A.2.10 (Small $R$).** For bounded $R \in (0, 2]$, the claim holds by decreasing $C_1$ so that $C_1 e^{\gamma R} \le 1 \le |\mathcal{P}_R|$; note that any ball contains at least one point, and the dependence on $R$ is monotone.

$\square$

**Lemma A.3** (Trees embed into hyperbolic space with constant distortion). *Let $T_n$ be any finite rooted $b$-ary tree of height $h$ and $n = \Theta(b^h)$ nodes with unit edge lengths (tree metric $d_T$). There exists an embedding $\iota : T_n \to \mathbb{H}_{c_H}^{d_H}$ such that for some constant $D_0 \geq 1$ (depending only on $b$, $d_H$, and $|c_H|$, but not on $n$ or $h$),*

$$\frac{1}{D_0}\, d_T(u,v) \;\leq\; d_{\mathbb{H}}\big(\iota(u), \iota(v)\big) \;\leq\; D_0\, d_T(u,v), \qquad \forall u, v \in T_n.$$

*Proof.* **Step A.3.0 (Normalization of curvature).** It suffices to construct $\iota$ into $\mathbb{H}_{-1}^d$ (curvature $-1$) with bi-Lipschitz distortion $D_0'(b,d)$. The general case follows by the standard rescaling isometry between $\mathbb{H}_{c_H}^{d_H}$ and $\mathbb{H}_{-1}^{d_H}$, which multiplies distances by $|c_H|^{-1/2}$ and alters only the constant by a factor depending on $|c_H|$ (Ratcliffe, 2006).

**Step A.3.1 (Radial–angular embedding scheme).** Fix a scale parameter $R > 0$ to be chosen later (depending only on $b, d$), and fix a root $o \in \mathbb{H}^d$. Each vertex at depth $t$ is embedded at radial distance $r(t) := Rt$ from $o$. To separate subtrees, assign to each vertex $v$ at depth $\ell$ a spherical cone $\mathsf{cone}(v) \subset \mathbb{S}^{d-1}$ of angular radius

$$\phi(\ell) \;=\; \phi_0\, e^{-R\ell},$$

for some $\phi_0 > 0$ independent of $n, h$. Children of $v$ receive disjoint sub-cones inside $\mathsf{cone}(v)$, each of radius at most $\phi(\ell+1)$. Such an allocation is possible provided

$$b \cdot \phi(\ell+1)^{d-1} \;\leq\; \tfrac{1}{2}\phi(\ell)^{d-1},$$

which (by substitution) holds whenever

$$R \;\geq\; \tfrac{1}{d-1}\, \log(2b). \tag{20}$$

This ensures both separation and angular slack at every level (cf. spherical cap packing arguments (Conway & Sloane, 2013)).

**Step A.3.2 (Angular separation across subtrees).** Let $u, v$ be two vertices with lowest common ancestor $w$ at depth $\ell$. Then the cones of different children of $w$ are disjoint, and hence

$$c_1\, e^{-R\ell} \;\leq\; \Theta(u,v) \;\leq\; c_2\, e^{-R\ell}, \tag{21}$$

for constants $c_1, c_2 > 0$ depending only on $(b,d)$. The lower bound comes from the enforced gap between sibling cones; the upper bound holds because each cone has radius $O(e^{-R\ell})$.

**Step A.3.3 (Hyperbolic law of cosines).** Write $s = \mathrm{depth}(u)$, $t = \mathrm{depth}(v)$, $r_s = Rs$, $r_t = Rt$, and $\ell = \mathrm{depth}(w)$. The hyperbolic law of cosines (curvature $-1$) gives

$$\cosh d_{\mathbb{H}}(\iota(u), \iota(v)) = \cosh r_s\, \cosh r_t - \sinh r_s\, \sinh r_t \cos \Theta(u,v), \tag{22}$$

see, e.g., (Anderson, 2006; Ratcliffe, 2006).

**Step A.3.4 (Lower bound).** Using $\cosh x \geq \tfrac{1}{2}e^x$ and $\sinh x \leq \tfrac{1}{2}e^x$, we deduce

$$\cosh d_{\mathbb{H}}(\iota(u), \iota(v)) \;\geq\; \tfrac{1}{4}e^{R(s+t)}\, (1 - \cos \Theta(u,v)).$$

By $1 - \cos\theta \geq \tfrac{\theta^2}{4}$ and Eq. (21),

$$\cosh d_{\mathbb{H}}(\iota(u), \iota(v)) \;\geq\; c \exp\big(R(s + t - 2\ell)\big).$$

Taking $\cosh^{-1}$ and using $\cosh^{-1}(y) \geq \log y$ for $y \geq 1$, we obtain

$$d_{\mathbb{H}}(\iota(u), \iota(v)) \;\geq\; R\, d_T(u,v) - C_1, \tag{23}$$

with $C_1 = C_1(b,d)$ independent of $n, h$.

**Step A.3.5 (Upper bound).** Construct a broken path: radially from $\iota(u)$ to $S(R\ell)$, then along $S(R\ell)$ by arc length $\sinh(R\ell)\Theta(u,v)$, then radially to $\iota(v)$. Thus

$$d_{\mathbb{H}}(\iota(u), \iota(v)) \;\leq\; R(s - \ell) + R(t - \ell) + \sinh(R\ell)\, \Theta(u,v).$$

By Eq. (21), $\sinh(R\ell)\Theta(u,v) \leq \tfrac{1}{2}e^{R\ell} \cdot c_2 e^{-R\ell} = O(1)$. Hence

$$d_{\mathbb{H}}(\iota(u), \iota(v)) \;\leq\; R\, d_T(u,v) + C_2. \tag{24}$$

**Step A.3.6 (Bi-Lipschitz conclusion).** Combining Eq. (23)–Eq. (24) gives

$$R\, d_T(u,v) - C_1 \;\leq\; d_{\mathbb{H}}(\iota(u),\iota(v)) \;\leq\; R\, d_T(u,v) + C_2.$$

Dividing by $d_T(u,v) \geq 1$ and adjusting constants yields

$$\frac{1}{D_0'}\, d_T(u,v) \;\leq\; d_{\mathbb{H}}(\iota(u),\iota(v)) \;\leq\; D_0'\, d_T(u,v),$$

for $D_0' = D_0'(b,d) < \infty$ independent of $n, h$.

**Step A.3.7 (Curvature rescaling).** Rescaling back to curvature $c_H < 0$ multiplies distances by $|c_H|^{-1/2}$, which only affects constants. Absorbing this factor into $D_0$ completes the proof. $\qquad\square$

**Lemma A.4** (Lower bounds for embedding trees into Euclidean/Hilbert). *Let $T_n$ be the vertex set of a rooted $b$-ary tree of height $h$ with unit edge lengths, so $n = \Theta(b^h)$. For any embedding $f : T_n \to \ell_2^m$, the bi-Lipschitz distortion satisfies*

$$\mathrm{dist}(f) \;\triangleq\; \Big(\sup_{u \neq v} \frac{\|f(u) - f(v)\|_2}{d_T(u,v)}\Big) \cdot \Big(\sup_{u \neq v} \frac{d_T(u,v)}{\|f(u) - f(v)\|_2}\Big) \;\geq\; c\,\sqrt{h} \;=\; c'\,\sqrt{\log n},$$

*for universal constants $c, c' > 0$ independent of $n, h$.*

*Proof.* We follow a standard Poincaré/energy argument for trees in Hilbert spaces, as developed in Linial et al. (1995) and refined in Gupta et al. (2004). Let $T$ denote a rooted $b$-ary tree of height $h$ with unit edge lengths, and let $V$, $E$, and $L$ be its sets of vertices, edges, and leaves, respectively. Then $|L| = \Theta(b^h)$ and $|E| = \Theta(|V|) = \Theta(n)$.

**Step A.4.1: Distortion, Lipschitz and co-Lipschitz constants.** For $f : V \to \ell_2^m$, write

$$\beta \;\triangleq\; \sup_{u \neq v} \frac{\|f(u) - f(v)\|_2}{d_T(u,v)} \qquad \text{and} \qquad \alpha \;\triangleq\; \sup_{u \neq v} \frac{d_T(u,v)}{\|f(u) - f(v)\|_2}.$$

Then the distortion is $\mathrm{dist}(f) = \beta \cdot \alpha$. Note that scaling $f$ by a positive constant scales $\beta$ and $\alpha$ inversely, leaving the product unchanged.

**Step A.4.2: Edge energy notation.** Root the tree at $o$. For each edge $e = (x, \mathrm{parent}(x)) \in E$ define the edge increment $a_e \;\triangleq\; f(x) - f(\mathrm{parent}(x)) \in \ell_2^m$. Then for any $v \in V$, if $P(o \to v)$ is the unique root-to-$v$ path,

$$f(v) - f(o) \;=\; \sum_{e \in P(o \to v)} a_e \qquad \text{(telescoping along the path).}$$

Let the (Hilbert) edge energy be

$$\mathcal{E}(f) \;\triangleq\; \sum_{e \in E} \|a_e\|_2^2.$$

By the definition of $\beta$ and unit edge lengths, $\|a_e\| \leq \beta$ for all $e$, hence

$$\mathcal{E}(f) \;=\; \sum_{e \in E} \|a_e\|^2 \;\leq\; |E| \cdot \beta^2. \tag{25}$$

**Step A.4.3: A tree Poincaré inequality for leaves (LLR).** Consider the multiset of ordered leaf pairs $L \times L$. Then there exists a universal constant $C > 0$ such that for every mapping $f : V \to \ell_2$,

$$\frac{1}{|L|^2} \sum_{u,v \in L} \|f(u) - f(v)\|_2^2 \;\leq\; C\, h \cdot \frac{1}{|E|} \sum_{e \in E} \|a_e\|_2^2. \tag{26}$$

Inequality Eq. (26) is a special case of the Poincaré type bounds in Linial et al. (1995) (see also Gupta et al. (2004) for related formulations), which control averaged pairwise squared distances on $L$ by the edge energy times the tree height.[1]

---

[1] Intuitively, $\|f(u) - f(v)\|^2$ can be expanded in terms of the increments $a_e$ along the two root-to-leaf paths, and averaging over all leaves makes cross-terms cancel while each level contributes additively; the factor $h$ accounts for the $h$ possible levels at which the two paths can diverge.

**Step A.4.4: Lower bound the LHS via co-Lipschitzness and leaf distances.** For leaves $u, v$, their tree distance is $d_T(u, v) = 2(h - \mathrm{depth}(\mathrm{LCA}(u, v)))$. If $U, V$ are drawn independently and uniformly from $L$, then $\mathbb{E}[d_T(U, V)] \geq c_0 h$ for some $c_0 = c_0(b) > 0$ (indeed, the LCA has bounded average depth). By Jensen's inequality, $\mathbb{E}[d_T(U, V)^2] \geq (\mathbb{E}[d_T(U, V)])^2 \geq c_0^2 h^2$. Using the co-Lipschitz property of $f$,

$$\|f(u) - f(v)\|_2 \geq \frac{1}{\alpha} d_T(u, v) \quad \Rightarrow \quad \|f(u) - f(v)\|_2^2 \geq \frac{1}{\alpha^2} d_T(u, v)^2.$$

Averaging over $u, v \in L$ yields

$$\frac{1}{|L|^2} \sum_{u,v \in L} \|f(u) - f(v)\|_2^2 \geq \frac{1}{\alpha^2} \cdot \frac{1}{|L|^2} \sum_{u,v \in L} d_T(u, v)^2 \geq \frac{c_0^2}{\alpha^2} h^2. \tag{27}$$

**Step A.4.5: Upper bound the RHS via the edge energy and $\beta$.** By Eq. (25), the RHS of Eq. (26) satisfies

$$C h \cdot \frac{1}{|E|} \sum_{e \in E} \|a_e\|_2^2 \leq C h \cdot \frac{1}{|E|} \cdot |E| \cdot \beta^2 = C h \beta^2. \tag{28}$$

**Step A.4.6: Combine the inequalities.** From Eq. (27) and Eq. (28), plugging into the Poincaré inequality Eq. (26), we obtain

$$\frac{c_0^2}{\alpha^2} h^2 \leq C h \beta^2.$$

Multiplying both sides by $\alpha^2$ yields

$$c_0^2 h^2 \leq C h \beta^2 \alpha^2.$$

Dividing by $h$ and taking square roots gives

$$\beta \alpha \geq \frac{c_0}{\sqrt{C}} \sqrt{h}.$$

Therefore, the distortion satisfies

$$\mathrm{dist}(f) = \beta \alpha \geq c \sqrt{h},$$

for some universal constant $c > 0$ independent of $n, h$.

**Step A.4.7: Express the bound in terms of $n$.** Since $n = \Theta(b^h)$, we have $h = \Theta(\log n)$. Consequently,

$$\mathrm{dist}(f) \geq c' \sqrt{\log n},$$

for a universal constant $c' > 0$.

$\square$

We now propagate these bounds from trees to hyperbolic balls.

**Lemma A.5** (Euclidean Lipschitz factor lower bounds on $B_{\mathbb{H}}(R)$). *Fix $R > 0$ and consider the hyperbolic ball $B_{\mathbb{H}}(R) \subset \mathbb{H}_{c_H}^{d_H}$. Let $f : B_{\mathbb{H}}(R) \to \ell_2^m$ be any mapping. There exist constants $C_1, \gamma > 0$ depending only on $d_H, |c_H|$ such that:*

*(A) Uniform separation (strong non-collapse). Assume there exists $S_R \subset B_{\mathbb{H}}(R)$ with $|S_R| \geq C_1 e^{\gamma R}$ such that $\|f(u) - f(v)\| \geq \Delta$ for all distinct $u, v \in S_R$, where $\Delta > 0$ is independent of $R$. Then for sufficiently large $R$,*

$$\beta_{\mathrm{euc}} \geq c_A \frac{e^{\gamma R/m}}{R},$$

*for some constant $c_A = c_A(d_H, |c_H|, m, \Delta) > 0$. In particular, since the exponential term dominates, there exists $c_A' > 0$ such that $\beta_{\mathrm{euc}} \geq c_A' R$ for all sufficiently large $R$.*

*(B) Global co-Lipschitz on a tree (moderate non-collapse). Assume there exists a rooted $b$-ary tree $T_n$ of height $h = \Theta(R)$ with $n = \Theta(e^{\gamma R})$, together with an embedding $\iota : T_n \to B_{\mathbb{H}}(R)$ of constant distortion (Lemma A.3), such that $f$ is co-Lipschitz on $\iota(T_n)$ with constant $\alpha_0 < \infty$, i.e., $\|f(\iota(u)) - f(\iota(v))\| \geq \alpha_0^{-1} d_T(u, v)$ for all $u, v \in T_n$. Then*

$$\beta_{\mathrm{euc}} \geq c_B \alpha_0^{-1} \sqrt{R},$$

*for some $c_B > 0$ depending only on $d_H, |c_H|$.*

*Proof.* We prove (A) and (B) separately. Throughout, $\beta_{\text{euc}}$ denotes the upper Lipschitz constant of $f$ with respect to the hyperbolic distance on $B_{\mathbb{H}}(R)$: $\|f(x) - f(y)\| \leq \beta_{\text{euc}} \, d_{\mathbb{H}}(x, y)$.

**Proof of (A): Euclidean packing vs. hyperbolic growth.** Let $o$ be the center of $B_{\mathbb{H}}(R)$. For any $x \in S_R \subset B_{\mathbb{H}}(R)$ we have $\|f(x) - f(o)\| \leq \beta_{\text{euc}} \, d_{\mathbb{H}}(x, o) \leq \beta_{\text{euc}} R$, so $f(S_R)$ lies in the Euclidean ball $B_{\ell_2^m}(f(o), \rho)$ with $\rho := \beta_{\text{euc}} R$. Since points in $S_R$ are $\Delta$-separated in $\ell_2^m$, the standard packing bound in $\mathbb{R}^m$ (e.g., (Papaspiliopoulos, 2020)) yields

$$|S_R| \;\leq\; \mathcal{N}_{\ell_2^m}(\rho, \Delta) \;\leq\; \left(1 + \frac{2\rho}{\Delta}\right)^m = \left(1 + \frac{2\beta_{\text{euc}} R}{\Delta}\right)^m.$$

Using $|S_R| \geq C_1 e^{\gamma R}$ by hypothesis, we obtain

$$C_1 e^{\gamma R} \;\leq\; \left(1 + \frac{2\beta_{\text{euc}} R}{\Delta}\right)^m \quad \Rightarrow \quad \beta_{\text{euc}} \;\geq\; \frac{\Delta}{2R}\left((C_1)^{1/m} e^{\gamma R/m} - 1\right).$$

Absorbing $(C_1)^{1/m}$ and the "$-1$" into a constant for large $R$ gives $\beta_{\text{euc}} \geq c_A \, e^{\gamma R/m}/R$. Since the exponential term dominates any polynomial, there exists $c_A' > 0$ such that $\beta_{\text{euc}} \geq c_A' R$ for all sufficiently large $R$.

**Proof of (B): Tree lower bound + co-Lipschitz control.** Let $F := f \circ \iota : T_n \to \ell_2^m$ be the composition with the tree embedding. By hypothesis, $F$ is co-Lipschitz on $T_n$ with constant $\alpha_0$, i.e., $\|F(u) - F(v)\| \geq \alpha_0^{-1} d_T(u, v)$ for all $u, v \in T_n$. Let

$$\beta_F \;:=\; \sup_{u \neq v} \frac{\|F(u) - F(v)\|}{d_T(u, v)}$$

be the upper Lipschitz constant of $F$ with respect to $d_T$. By the Linial–London–Rabinovich/Gupta–Newman–Rabinovich–Sinclair lower bound for trees into Hilbert spaces ((Linial et al., 1995); see also (Gupta et al., 2004)), the bi-Lipschitz distortion of any $F : T_n \to \ell_2$ satisfies

$$\text{dist}(F) = \beta_F \, \alpha_F \;\geq\; c_* \sqrt{h},$$

where $\alpha_F$ is the co-Lipschitz constant of $F$ and $c_* > 0$ is universal. Since $\alpha_F \leq \alpha_0$, we get

$$\beta_F \;\geq\; \frac{c_*}{\alpha_0} \sqrt{h}.$$

Next, relate $\beta_F$ to $\beta_{\text{euc}}$. By Lemma A.3, $\iota$ has constant distortion $D_0$, so for all $u \neq v$

$$\|F(u) - F(v)\| \;=\; \|f(\iota(u)) - f(\iota(v))\| \;\leq\; \beta_{\text{euc}} \, d_{\mathbb{H}}(\iota(u), \iota(v)) \;\leq\; \beta_{\text{euc}} \, D_0 \, d_T(u, v),$$

hence $\beta_F \leq D_0 \, \beta_{\text{euc}}$. Combining the last two displays and using $h = \Theta(R)$ from the tree–ball construction,

$$\beta_{\text{euc}} \;\geq\; \frac{1}{D_0} \beta_F \;\geq\; \frac{c_*}{D_0 \, \alpha_0} \sqrt{h} \;=\; c_B \, \alpha_0^{-1} \sqrt{R},$$

with $c_B > 0$ depending only on $d_H, |c_H|$ through $D_0$ and the height–radius comparability. $\qquad\square$

*Remark (Geometric penalties of Euclidean embeddings).* Lemmas A.2–A.5 collectively demonstrate that hyperbolic components of data manifolds impose unavoidable penalties when represented in purely Euclidean spaces:

In the *hyperbolic regime*, exponential packing growth (Lemma A.2) together with tree–hyperbolic embeddings (Lemma A.3) and the Euclidean distortion lower bound for trees (Lemma A.4) imply that any Euclidean embedding of $B_{\mathbb{H}}(R)$ must either inflate the Lipschitz constant to at least $\Omega(R)$ or absorb an additive residual $\varepsilon_{\text{dist,euc}} = \Omega(R)$.

In the *tree co-Lipschitz regime*, the GNRS/LLR bounds show that co-Lipschitz control still forces $\beta_{\text{euc}} \geq \Omega(\sqrt{R})$, so the geometric residual cannot be smaller than $\Omega(\sqrt{R})$.

These findings confirm that the geometric term $\varepsilon_{\text{dist}}$ in Theorem 4.1 necessarily deteriorates with $R$ when hyperbolic factors are present. They also provide the foundation for Theorem 4.2, where product spaces are shown to mitigate such distortions.

Finally, in addition to hyperbolic penalties, spherical factors introduce a distinct and *fixed* gap: Euclidean embeddings replace geodesic arcs by chords. As we establish next (Lemma A.6), this substitution inevitably incurs a bi-Lipschitz distortion of at least $\pi/2$, independent of the radius. Hence, while hyperbolic geometry amplifies distortion with scale, spherical geometry contributes an irreducible constant distortion gap.

### A.2.2 SPHERICAL FACTOR: CHORD VS. ARC AND A FIXED GAP

We next quantify the irreducible loss of using Euclidean chordal distance in place of spherical geodesics.

**Lemma A.6** (Spherical arc–chord inequality and sharp bi-Lipschitz constants). *Let $\mathbb{S}_{R_S}^{d_S} \subset \mathbb{R}^{d_S+1}$ be the $d_S$-dimensional sphere of radius $R_S > 0$ with the induced Euclidean metric $\|\cdot\|_2$ from the ambient space. For any two points $p, q \in \mathbb{S}_{R_S}^{d_S}$, let $\theta = \theta(p, q) \in [0, \pi]$ denote the central angle at the sphere's center between $p$ and $q$. Then the spherical geodesic (arc) distance and the chordal (ambient Euclidean) distance satisfy*

$$d_{\mathrm{geo}}(p, q) = R_S\,\theta, \qquad d_{\mathrm{chord}}(p, q) = \|p - q\|_2 = 2R_S\,\sin(\theta/2),$$

*and the following double inequality holds:*

$$\frac{2}{\pi}\,d_{\mathrm{geo}}(p, q) \;\le\; d_{\mathrm{chord}}(p, q) \;\le\; d_{\mathrm{geo}}(p, q). \tag{29}$$

*Consequently, the identity map*

$$\mathrm{id} : \left(\mathbb{S}_{R_S}^{d_S}, d_{\mathrm{geo}}\right) \longrightarrow \left(\mathbb{S}_{R_S}^{d_S}, d_{\mathrm{chord}}\right)$$

*is bi-Lipschitz with Lipschitz constant 1 and co-Lipschitz constant $2/\pi$ (equivalently, its inverse has Lipschitz constant $\pi/2$). Therefore the distortion is exactly $\pi/2$, and these constants are sharp (Burago et al., 2001; Berger, 2009; Bridson & Haefliger, 2013).*

*Proof.* **Step A.6.1 (Formulas for $d_{\mathrm{geo}}$ and $d_{\mathrm{chord}}$).** By definition of the spherical metric on a sphere of radius $R_S$, the length of a minimizing geodesic (great-circle arc; unique unless $\theta = \pi$) between $p$ and $q$ equals

$$d_{\mathrm{geo}}(p, q) = R_S\,\theta,$$

where $\theta \in [0, \pi]$ is the central angle between $p$ and $q$ (Berger, 2009). Since $p$ and $q$ lie on the sphere of radius $R_S$ in $\mathbb{R}^{d_S+1}$, the ambient Euclidean (chordal) distance is

$$d_{\mathrm{chord}}(p, q) = \|p - q\|_2 = \sqrt{\|p\|_2^2 + \|q\|_2^2 - 2\langle p, q\rangle} = \sqrt{2R_S^2\big(1 - \cos\theta\big)}.$$

Using the identity $1 - \cos\theta = 2\sin^2(\theta/2)$, we obtain

$$d_{\mathrm{chord}}(p, q) = 2R_S\,\sin(\theta/2). \tag{30}$$

**Step A.6.2 (Upper bound $d_{\mathrm{chord}} \le d_{\mathrm{geo}}$).** For all $x \ge 0$ we have $\sin x \le x$. Apply this with $x = \theta/2 \in [0, \pi/2]$:

$$\sin(\theta/2) \;\le\; \theta/2.$$

Multiplying by $2R_S$ and invoking Eq. (30) gives

$$d_{\mathrm{chord}}(p, q) = 2R_S\,\sin(\theta/2) \;\le\; R_S\,\theta = d_{\mathrm{geo}}(p, q).$$

**Step A.6.3 (Lower bound $d_{\mathrm{chord}} \ge \frac{2}{\pi}d_{\mathrm{geo}}$).** Consider $\phi(x) := \sin x - \frac{2}{\pi}x$ on $[0, \pi/2]$. We have $\phi(0) = \phi(\pi/2) = 0$ and $\phi''(x) = -\sin x \le 0$, so $\phi$ is concave. A concave function lies above its chord, i.e., $\sin x \ge \frac{2}{\pi}x$ for all $x \in [0, \pi/2]$. Taking $x = \theta/2$ yields

$$\sin(\theta/2) \;\ge\; \frac{\theta}{\pi}.$$

Multiplying by $2R_S$ gives

$$d_{\mathrm{chord}}(p, q) = 2R_S\,\sin(\theta/2) \;\ge\; \frac{2}{\pi}R_S\,\theta = \frac{2}{\pi}d_{\mathrm{geo}}(p, q).$$

**Step A.6.4 (Bi-Lipschitz constants and sharpness).** The inequalities Eq. (29) are equivalent to

$$\frac{2}{\pi}\,d_{\mathrm{geo}}(p, q) \;\le\; d_{\mathrm{chord}}(p, q) \;\le\; d_{\mathrm{geo}}(p, q) \quad \text{for all } p, q \in \mathbb{S}_{R_S}^{d_S}.$$

Hence for the identity $f = \mathrm{id}$ we have the (global) Lipschitz seminorm

$$L(f) \ := \ \sup_{p \neq q} \frac{d_{\mathrm{chord}}(p, q)}{d_{\mathrm{geo}}(p, q)} \ = \ 1,$$

and the co-Lipschitz constant

$$\ell(f) \ := \ \inf_{p \neq q} \frac{d_{\mathrm{chord}}(p, q)}{d_{\mathrm{geo}}(p, q)} \ = \ \frac{2}{\pi}.$$

Therefore the metric distortion is

$$\mathrm{Dist}(f) \ = \ \frac{L(f)}{\ell(f)} = \frac{\pi}{2}.$$

Sharpness: as $\theta \to 0$, $\frac{d_{\mathrm{chord}}}{d_{\mathrm{geo}}} \to 1$ so $L(f) = 1$; at $\theta = \pi$, $\frac{d_{\mathrm{chord}}}{d_{\mathrm{geo}}} = \frac{2}{\pi}$ so $\ell(f) = 2/\pi$. Thus $\mathrm{Dist}(f) = \pi/2$ is attained, proving the constants are optimal.

**Step A.6.5 (Consequence).** Any Euclidean-only representation that relies on chordal distances for spherical data incurs an irreducible $\pi/2$ distortion relative to the intrinsic spherical metric (Burago et al., 2001). By contrast, using spherical geometry yields $(L(f), \ell(f)) = (1, 1)$ up to negligible projection/discretization error. □

### A.2.3 FLAT FACTOR: NO ADVANTAGE OR DISADVANTAGE

We now analyze the Euclidean (flat) component of the product manifold. Unlike hyperbolic and spherical components, Euclidean geometry introduces no additional distortion because geodesic distances coincide exactly with the standard $\ell_2$ metric.

**Lemma A.7** (Flat factor is preserved without distortion). *Let* $(\mathbb{E}^{d_E}, \|\cdot\|_2)$ *denote* $d_E$-*dimensional Euclidean space with its canonical norm. Then for any two points* $u, v \in \mathbb{E}^{d_E}$:

$$d_{\mathrm{geo}}(u, v) = \|u - v\|_2,$$

*where* $d_{\mathrm{geo}}$ *is the geodesic distance induced by the Euclidean metric. Consequently, any embedding* $\iota : (\mathbb{E}^{d_E}, \|\cdot\|_2) \to \ell_2^m$ *that is an isometric inclusion satisfies*

$$\frac{1}{1}\|u - v\|_2 \ \leq \ \|\iota(u) - \iota(v)\|_2 \ \leq \ 1 \cdot \|u - v\|_2,$$

*so the bi-Lipschitz distortion is exactly* 1.

*Proof.* **Step A.7.1 (Definition of geodesic distance).** On any Riemannian manifold $(\mathcal{M}, g)$, the geodesic distance between two points is the infimum of lengths of smooth curves $\gamma : [0, 1] \to \mathcal{M}$ with $\gamma(0) = u, \gamma(1) = v$:

$$d_{\mathrm{geo}}(u, v) \ = \ \inf_{\gamma} \int_0^1 \sqrt{g_{\gamma(t)}(\dot{\gamma}(t), \dot{\gamma}(t))} \, dt.$$

See, e.g., Do Carmo & Flaherty Francis (1992) or Lee (2006).

**Step A.7.2 (Euclidean case).** For $\mathcal{M} = \mathbb{E}^{d_E}$ with $g = \langle \cdot, \cdot \rangle$ the standard Euclidean inner product, the straight line

$$\gamma(t) = (1 - t)u + tv, \quad t \in [0, 1],$$

is a (globally) minimizing geodesic; see Do Carmo & Flaherty Francis (1992). Its derivative is constant $\dot{\gamma}(t) = v - u$, hence

$$\int_0^1 \sqrt{\langle v - u, v - u \rangle} \, dt = \int_0^1 \|v - u\|_2 \, dt = \|v - u\|_2,$$

which gives $d_{\mathrm{geo}}(u, v) \leq \|v - u\|_2$.

*(Lower bound for any curve).* For any absolutely continuous $\gamma$ with $\gamma(0) = u, \gamma(1) = v$, the triangle inequality for vector integrals (Minkowski) yields

$$\int_0^1 \|\dot{\gamma}(t)\|_2 \, dt \ \geq \ \left\| \int_0^1 \dot{\gamma}(t) \, dt \right\| = \|\gamma(1) - \gamma(0)\|_2 \ = \ \|v - u\|_2,$$

see, e.g., Burago et al. (2001). Taking infimum over all such $\gamma$ gives $d_{\text{geo}}(u, v) \geq \|v - u\|_2$. Combining both inequalities,

$$d_{\text{geo}}(u, v) = \|u - v\|_2.$$

**Step A.7.3 (Bi-Lipschitz constants).** Consider an isometric inclusion $\iota : \mathbb{E}^{d_E} \hookrightarrow \ell_2^m$ (e.g., $\iota(x) = (x, 0, \ldots, 0)$). Then

$$\|\iota(u) - \iota(v)\|_2 = \|u - v\|_2 = d_{\text{geo}}(u, v),$$

so the Lipschitz and co-Lipschitz constants are both 1, and the bi-Lipschitz distortion equals 1 (Burago et al., 2001).

**Step A.7.4 (Conclusion).** The distortion $\beta/\alpha = 1$ exactly, which is the minimum possible. Therefore, unlike the hyperbolic or spherical factors, the flat factor introduces no geometric mismatch between Euclidean-only and product spaces. $\qquad\square$

This lemma simply formalizes the fact that in flat Euclidean geometry, geodesic and Euclidean distances coincide, so both Euclidean-only and product manifolds preserve distances identically for the flat component. Thus there is no advantage or disadvantage on this factor.

### A.2.4 COMBINING FACTORS: PRODUCT VS. EUCLIDEAN

We now combine the factor-wise analyses of hyperbolic, spherical, and Euclidean components to compare the geometric distortion parameters $\beta(\kappa)$ and $\varepsilon_{\text{dist}}(\kappa)$ between a purely Euclidean embedding and a curvature-aware product embedding.

**Lemma A.8** (Comparison of distortion: Euclidean-only vs. product space)**.** *Let the data manifold satisfy mixed-curvature product manifold, i.e., $\mathcal{M}^\star \subset \mathbb{E}^{d_E} \times \mathbb{H}_{c_H}^{d_H} \times \mathbb{S}_{c_S}^{d_S}$. Denote by $\beta_{\text{euc}}, \varepsilon_{\text{dist,euc}}$ the distortion constants for an embedding into Euclidean space $\ell_2^m$, and by $\beta_{\text{prod}}, \varepsilon_{\text{dist,prod}}$ those for the natural product space embedding. Then there exist positive constants $c_1, c_2$ (depending only on curvature parameters and the geodesic radius $R$) such that for the Euclidean-only embedding at least one of the following holds:*

$$\beta_{\text{euc}} \geq 1 + c_1\sqrt{R} \qquad or \qquad \varepsilon_{\text{dist,euc}} \geq \varepsilon_{\text{dist,prod}} + c_2\sqrt{R}.$$

*If the embedding additionally preserves uniform separation of exponentially many points (the strong-separation regime), the bound strengthens to*

$$\beta_{\text{euc}} \geq 1 + c_1 R \qquad or \qquad \varepsilon_{\text{dist,euc}} \geq \varepsilon_{\text{dist,prod}} + c_2 R.$$

*In contrast, for the product space embedding one has*

$$\beta_{\text{prod}} = 1, \qquad \varepsilon_{\text{dist,prod}} \leq \varepsilon_{\text{num}},$$

*where $\varepsilon_{\text{num}}$ denotes negligible numerical residuals.*

*Proof.* **Step A.8.1 (Hyperbolic factor).** By Lemma A.2 and Lemma A.3, hyperbolic balls contain exponentially many well-separated points and large embedded trees of height $\Theta(R)$. Lemma A.5 and Lemma A.4 then imply that any Euclidean embedding must either incur multiplicative distortion

$$\beta_{\text{euc}}^{(\mathbb{H})} \geq c_H\sqrt{R}$$

(or $\beta_{\text{euc}}^{(\mathbb{H})} \gtrsim R$ in the strong-separation regime), or else pay additive error of the same order. By contrast, the product space contains a hyperbolic factor directly, achieving $\beta_{\text{prod}}^{(\mathbb{H})} = 1$ and $\varepsilon_{\text{dist,prod}}^{(\mathbb{H})} = 0$ up to numerical tolerance.

**Step A.8.2 (Spherical factor).** By Lemma A.6, the identity map from $(\mathbb{S}_{R_S}^{d_S}, d_{\text{geo}})$ to chordal distances in $\ell_2$ has distortion exactly $\pi/2$ (Burago et al., 2001; Berger, 2009; Bridson & Haefliger, 2013). Thus any Euclidean-only embedding suffers a fixed constant gap: either the Lipschitz constant $\beta_{\text{euc}}^{(\mathbb{S})}$ is bounded away from 1, or the co-Lipschitz constant $\alpha_{\text{euc}}^{(\mathbb{S})}$ falls below 1. This structural distortion cannot be removed by uniform rescaling. In the product embedding, the spherical factor is included, yielding $\beta_{\text{prod}}^{(\mathbb{S})} = \alpha_{\text{prod}}^{(\mathbb{S})} = 1$.

**Step A.8.3 (Flat factor).** Lemma A.7 shows that in Euclidean components the geodesic and ambient distances coincide, so both Euclidean-only and product embeddings preserve distances exactly.

**Step A.8.4 (Aggregation via product metric).** For product manifolds under the $\ell_2$ product metric, the overall Lipschitz constant is the maximum of the factor-wise constants, while the additive residual is at most the $\ell_2$-sum (hence also bounded by the $\ell_1$-sum) of the factor-wise residuals (Burago et al., 2001). Therefore,

$$\beta_{\mathrm{prod}} = \max\{1, 1, 1\} = 1, \qquad \varepsilon_{\mathrm{dist,prod}} \leq \varepsilon_{\mathrm{num}}.$$

For Euclidean-only embeddings, however, the hyperbolic factor enforces $\beta_{\mathrm{euc}} \gtrsim \sqrt{R}$ (or $R$) unless additive errors of the same order are allowed, and the spherical factor enforces a constant distortion gap of at least $\pi/2$. Combining these yields the claimed inequalities. $\qquad\square$

This lemma formalizes that Euclidean-only embeddings cannot simultaneously capture exponential hyperbolic growth and spherical angular structure: at least one distortion parameter must diverge with $R$ or remain bounded away from zero. In contrast, the curvature-aware product space accommodates all three geometries natively, achieving unit multiplicative constants and negligible residuals. This factor-wise advantage underlies the tighter bounds of Theorem 4.2.

A.2.5    PROOF OF THEOREM 4.2

We compare the general bound Theorem 4.1 under two matching spaces: Euclidean-only (subscript "euc") and the curvature-aware product space (subscript "prod").

**Step Theorem 4.2.1: Write both bounds explicitly:**

$$\Delta_{\mathrm{euc}} \ \leq \ L\Big(\beta_{\mathrm{euc}}\big[\overline{\mathcal{E}}_{\mathrm{stat}}(M) + \varepsilon_{\mathrm{opt}}\big] \ + \ C\,\varepsilon_{\mathrm{dist,euc}}\Big) \ + \ \varepsilon_{\mathrm{stab}},$$

$$\Delta_{\mathrm{prod}} \ \leq \ L\Big(\beta_{\mathrm{prod}}\big[\overline{\mathcal{E}}_{\mathrm{stat}}(M) + \varepsilon_{\mathrm{opt}}\big] \ + \ C\,\varepsilon_{\mathrm{dist,prod}}\Big) \ + \ \varepsilon_{\mathrm{stab}}.$$

For brevity, denote $S := \overline{\mathcal{E}}_{\mathrm{stat}}(M) + \varepsilon_{\mathrm{opt}} > 0$.

**Step Theorem 4.2.2: Product-space geometry (factor-wise aggregation).** By Lemma A.7 (flat factor), Lemma A.6 (spherical factor with geodesic metric), and the construction of the hyperbolic factor (isometric up to negligible projection), the product space achieves

$$\beta_{\mathrm{prod}} = 1, \qquad \varepsilon_{\mathrm{dist,prod}} \leq \varepsilon_{\mathrm{num}},$$

where $\varepsilon_{\mathrm{num}} \geq 0$ collects small numerical/projection errors. This uses the $\ell_2$ product aggregation; the overall Lipschitz constant equals the *maximum* of factor-wise constants, and the additive deviation is at most the $\ell_2$-sum of factor-wise deviations (hence $\leq$ the $\ell_1$-sum) (Burago et al., 2001).

**Step Theorem 4.2.3: Euclidean-only geometry — hyperbolic and spherical factors.** We now lower bound the Euclidean-only distortion using our factor-wise lemmas.

*(Hyperbolic factor).* By Lemma A.2 and Lemma A.3, hyperbolic balls contain exponentially many well-separated points and constant-distortion embedded trees of height $\Theta(R)$. Lemma A.5 and Lemma A.4 then imply that any Euclidean embedding must either incur multiplicative distortion

$$\beta_{\mathrm{euc}}^{(\mathbb{H})} \ \geq \ c_A\,\frac{e^{\gamma R/m}}{R} \quad (\text{hence } \beta_{\mathrm{euc}}^{(\mathbb{H})} \geq \Omega(R) \text{ for large } R)$$

under uniform separation (packing in $\ell_2^m$; see Vershynin, 2009), or else pay additive residual $\varepsilon_{\mathrm{dist,euc}}^{(\mathbb{H})} \geq c_A' R$ of the same order (refined multiplicative–additive trade-off). If $f$ is only globally co-Lipschitz on an embedded $b$-ary tree of height $h = \Theta(R)$, then combining tree lower bounds into Hilbert space (e.g., complete binary trees have Euclidean distortion $\Theta(\sqrt{h})$) yields

$$\beta_{\mathrm{euc}}^{(\mathbb{H})} \ \geq \ c_B\,\alpha_0^{-1}\sqrt{R},$$

and enforcing $\beta_{\mathrm{euc}}^{(\mathbb{H})} = o(\sqrt{R})$ forces $\varepsilon_{\mathrm{dist,euc}}^{(\mathbb{H})} \geq c_B'\sqrt{R}$ along $\Theta(n)$ root-to-leaf paths (Linial & Saks, 2003; Matoušek, 1999).

*(Spherical factor).* By Lemma A.6 (arc–chord inequality),

$$\frac{\beta_{\mathrm{euc}}^{(\mathbb{S})}}{\alpha_{\mathrm{euc}}^{(\mathbb{S})}} \ \geq \ \frac{\pi}{2},$$

so any Euclidean-only representation relying on chordal distances for intrinsically spherical data incurs a fixed bi-Lipschitz gap: either $\beta_{\text{euc}}^{(\mathbb{S})} \geq 1 + c_S$ or $\alpha_{\text{euc}}^{(\mathbb{S})} \leq 1 - c'_S$ for constants $c_S, c'_S > 0$ (Burago et al., 2001; Berger, 2009; Bridson & Haefliger, 2013). This gap cannot be removed by uniform rescaling without worsening other factors.

**Step Theorem 4.2.4: From factor-wise to overall Euclidean-only bounds.** Under the $\ell_2$ product aggregation,

$$\beta_{\text{euc}} \geq \max\left\{\beta_{\text{euc}}^{(\mathbb{H})}, \beta_{\text{euc}}^{(\mathbb{S})}, 1\right\}, \qquad \varepsilon_{\text{dist,euc}} \geq \max\left\{\varepsilon_{\text{dist,euc}}^{(\mathbb{H})}, \varepsilon_{\text{dist,euc}}^{(\mathbb{S})}\right\}$$

(the latter follows by taking point pairs supported on the single worst factor). Therefore, in the strong non-collapse regime we must have either $\beta_{\text{euc}} \geq 1 + c_1 R$ or $\varepsilon_{\text{dist,euc}} \geq \varepsilon_{\text{dist,prod}} + c_2 R$, and under global co-Lipschitzness either $\beta_{\text{euc}} \geq 1 + c_1\sqrt{R}$ or $\varepsilon_{\text{dist,euc}} \geq \varepsilon_{\text{dist,prod}} + c_2\sqrt{R}$, with an additional constant spherical gap in both cases.

**Step Theorem 4.2.5: Compare the *bounds*.** Let

$$U_{\text{euc}} := L\big(\beta_{\text{euc}} S + C\,\varepsilon_{\text{dist,euc}}\big) + \varepsilon_{\text{stab}}, \qquad U_{\text{prod}} := L\big(\beta_{\text{prod}} S + C\,\varepsilon_{\text{dist,prod}}\big) + \varepsilon_{\text{stab}}.$$

Then

$$U_{\text{euc}} - U_{\text{prod}} = L\Big((\beta_{\text{euc}} - \beta_{\text{prod}})\,S + C\big(\varepsilon_{\text{dist,euc}} - \varepsilon_{\text{dist,prod}}\big)\Big).$$

Using Step 2 ($\beta_{\text{prod}} = 1$, $\varepsilon_{\text{dist,prod}} \leq \varepsilon_{\text{num}}$) and Step 4,

$$U_{\text{euc}} - U_{\text{prod}} \geq \begin{cases} L\big(a_1 R\,S + Ca_2 R\big) & \text{(strong non-collapse)}, \\ L\big(b_1\sqrt{R}\,S + Cb_2\sqrt{R}\big) & \text{(global co-Lipschitz)}, \end{cases}$$

for some $a_1, a_2, b_1, b_2 > 0$ depending on curvature and non-collapse constants, with the spherical factor contributing an additional constant gap to the additive side.

**Step Theorem 4.2.6: Conclude Theorem 4.2.** Since $\Delta_{\text{euc}} \leq U_{\text{euc}}$ and $\Delta_{\text{prod}} \leq U_{\text{prod}}$, we obtain a strictly tighter guarantee for product-space matching:

$$\Delta_{\text{prod}} \leq \Delta_{\text{euc}} - L\,\delta, \qquad \delta \in \big\{a_1 R\,S + Ca_2 R, \ b_1\sqrt{R}\,S + Cb_2\sqrt{R}\big\} > 0.$$

Hence, whenever the real data manifold contains nontrivial hyperbolic or spherical components, curvature-aware product matching strictly improves the bound in Theorem 4.1 For the MMD variant, replace the IPM dual step by Gretton et al. (2012); the metric lower bounds (Steps 3–4) are purely geometric and remain unchanged.

**Remark A.9** (Geometric intuition for Theorem 4.2). *The theorem highlights a simple but fundamental fact: when the underlying data manifold contains hyperbolic or spherical components, Euclidean space is inherently mismatched. In the hyperbolic case, exponential volume growth forces either a multiplicative Lipschitz constant that diverges with radius $R$ or an additive residual of the same order (via packing and tree-embedding lower bounds). In the spherical case, the arc–chord inequality guarantees an irreducible $\pi/2$ bi-Lipschitz gap (Burago et al., 2001; Berger, 2009). By contrast, the curvature-aware product space faithfully incorporates Euclidean, hyperbolic, and spherical factors, so that each component achieves distortion 1 (up to negligible projection error). As a result, the product embedding yields strictly tighter error guarantees in Theorem 4.2, whereas a Euclidean-only embedding must inevitably pay a price that scales with $R$ or persists as a constant gap.*

# B    USAGE OF LLM

In this work, we utilized ChatGPT, an AI language model developed by OpenAI, to assist with the following tasks:

- **Code**: During the coding stage, ChatGPT-4o was used to assist in debugging by providing suggestions for correcting coding errors. All implementations were ultimately written, tested, and verified by the author.

---

**Algorithm 1:** GeoDM - Dataset Distillation in Product Space with Geometry-Aware Matching

---

**Input:** Real dataset $\mathcal{D}_r = \{x_i\}_{i=1}^N$, number of synthetic samples $M$, product space
$\qquad \mathcal{P} = \mathbb{E}^{d_E} \times \mathbb{H}_{c_H}^{d_H} \times \mathbb{S}_{c_S}^{d_S}$, learning rate $\eta$, max iterations $T$

**Output:** Synthetic dataset $\mathcal{D}_s = \{\tilde{x}_j\}_{j=1}^M$

**Initialization:** Initialize synthetic data $\mathcal{D}_s$, learnable curvatures $c_H < 0$, $c_S > 0$, and learnable
weights $w_E, w_H, w_S$;

**for** $t = 1$ **to** $T$ **do**
    Sample minibatches $X_r \subset \mathcal{D}_r$, $X_s \subset \mathcal{D}_s$;
    **for** $x \in X_r \cup X_s$ **do**
        Extract Euclidean features $f_E(x)$ via CNN;
        Extract Hyperbolic features $f_H(x) = \exp_0^{c_H}(\text{CNN}(x))$ via Riemannian CNN;
        Extract Spherical features $f_S(x)$ via spherical CNN;
        Form embedding $z(x) = (f_E(x), f_H(x), f_S(x)) \in \mathcal{P}$;
    Compute distribution matching using $z(x)$ with Eq. (2) ;
    Compute optimal transport loss with Eq. (6) ;
    Compute learnable curvature with Eq. (3) ;
    Compute learnable weight with Eq. (5) ;
    Overall loss $\mathcal{L}_{\text{total}} \leftarrow \mathcal{L}_{\text{DM}} + \lambda_{\text{OT}}\mathcal{L}_{\text{OT}} + \lambda_c\mathcal{L}_{\text{curv}}$;
    Update $\mathcal{D}_s \leftarrow \mathcal{D}_s - \eta\nabla_{\mathcal{D}_s}\mathcal{L}_{\text{total}}$;

---

- **Writing**: ChatGPT-4o was used exclusively during the final stage of manuscript preparation to assist in polishing the English descriptions within the experimental analysis section. All generated suggestions were carefully reviewed, edited, and adapted by the authors to ensure technical accuracy and align with the original intent. Other sections of the paper were not generated or rewritten using AI tools.

No AI tool was used to generate core content or conclusions.

## C  ALGORITHM

The pseudo code of our GeoDM algorithm is described in Algorithm 1.

## D  SYNTHETIC IMAGES

Condensed images in Figure 4.

## E  HYPERPARAMETER SETTING

We adopt a Convnet-3 backbone with batch normalization and width multiplier 1.0. Synthetic training uses a batch size of 128. Training runs for 1500 evaluation epochs with evaluation every 100 epochs, preceded by 60 pretraining epochs. Optimization is performed using AdamW with base learning rate 0.001, momentum 0.9, and weight decay $5 \times 10^{-4}$. For synthetic images, we use learning rate 0.01 and momentum 0.5. The overall condensation process runs for 10,000 iterations with frequency parameter 4096 and factor 2. Data augmentation includes mixup (Beta = 1.0, probability 0.5). For curvature regularization, we fix $\lambda_H = \lambda_S = 1$. The learnable geometry weights for Euclidean, hyperbolic, and spherical components are initialized uniformly at $1/3$ each. All experiments are run with 3 independent model initializations and random seed 0.

## F  RELATED WORK

**Dataset distillation with distribution matching and feature matching**  Dataset distillation (DD) was first introduced by (Wang et al., 2018) with the aim of synthesizing compact datasets that enable models trained on it to achieve performance comparable to those trained on the original large

1404
1405
1406
1407
1408
1409
1410
1411
1412
1413
1414
1415
1416
1417
1418
1419
1420
1421
1422
1423
1424
1425
1426
1427
1428
1429
1430
1431
1432
1433
1434
1435
1436
1437
1438
1439
1440
1441
1442
1443
1444
1445
1446
1447
1448
1449
1450
1451
1452
1453
1454
1455
1456
1457

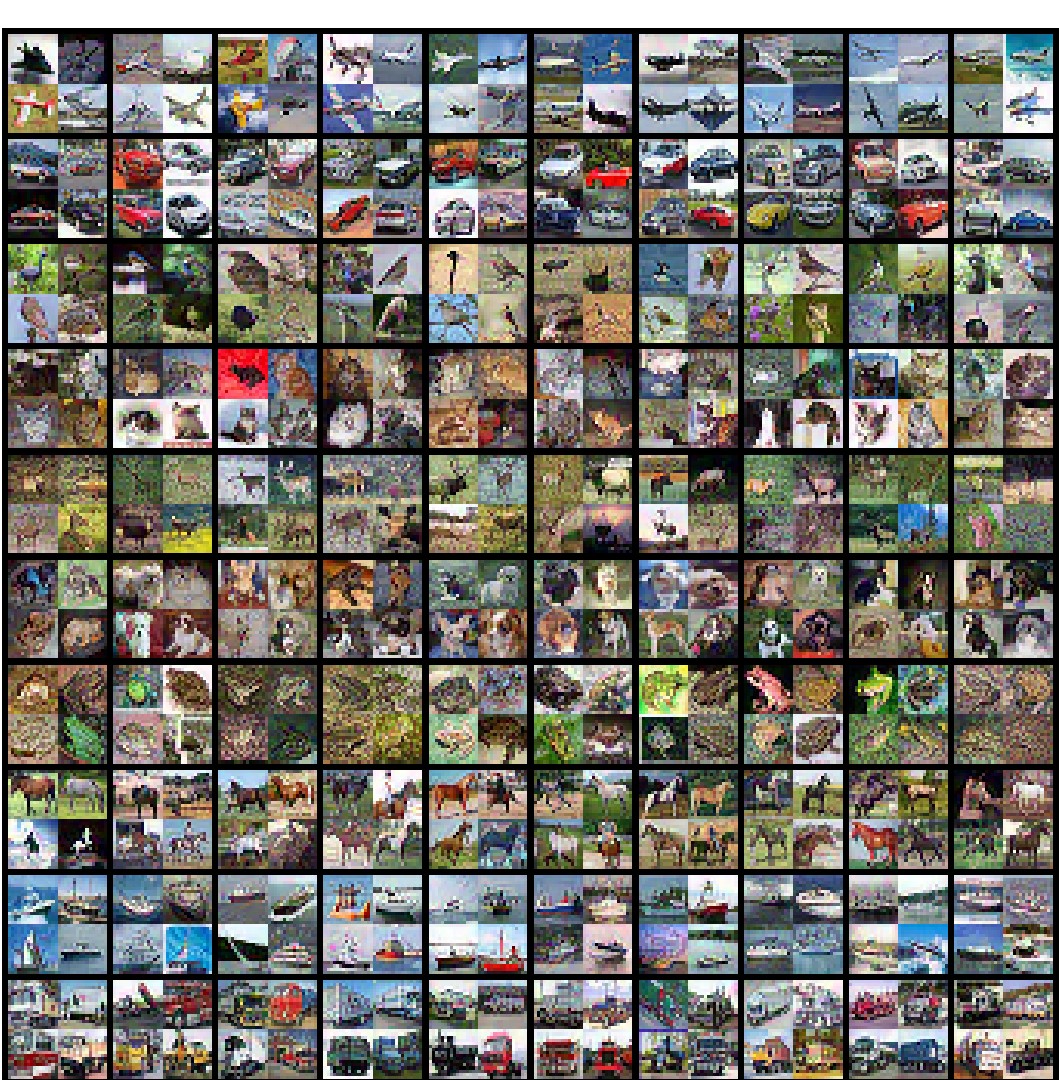

Figure 4: Condensed images

dataset. Among various approaches, distribution matching (DM) (Zhao & Bilen, 2023) has been recognized as a simple yet efficient framework that maintains a good balance between accuracy and efficiency. Within the DM family, CAFE (Wang et al., 2022) proposed a curriculum-based strategy to progressively match features from shallow to deep layers. M3D (Zhang et al., 2024) minimized discrepancies in higher-order moments to strengthen distribution-level consistency, while DSDM (Li et al., 2024) further advanced this by conducting distribution matching directly in the feature space with a scalable optimization strategy. WDDD (Liu et al., 2023a) employs Wasserstein metrics for feature-level matching, while OPTICAL (Cui et al., 2025) leverages optimal transport to allocate contributions during dataset distillation. More recently, NCFM (Wang et al., 2025) proposed a novel condensation framework that captures the overall distributional discrepancy between synthetic and real data. While Li et al. (2025) use hyperbolic space to do the distribution matching, it doesn't declare the advantage of hyperbolic space. And single hyperbolic space will not surpass Euclidean space in most of the situation.

**Manifold learning with non-Euclidean and product geometries**   A growing body of work emphasizes that the geometry of the representation space should match the structural biases of data (Nickel & Kiela, 2017; Meila & Zhang, 2023). Hyperbolic models (Ganea et al., 2018) are effective in capturing hierarchical growth, with extensions to embeddings, networks, and graph convolutions, often incorporating trainable curvature to enhance flexibility (Wang et al., 2021). Spherical models (Davidson et al., 2018) handle directional or rotation-equivariant signals by employing hyperspherical latent variables and spherical convolutions. To better represent heterogeneous structures, product manifolds that combine Euclidean, spherical, and hyperbolic components have been proposed (Gu et al., 2018), offering a principled way to encode mixed curvatures. Recent advances extend this line by introducing mixed-curvature spaces (Wang et al., 2021) where curvature parameters are learned jointly with embeddings, thereby adapting geometry to data in a more flexible manner. Beyond embeddings, general-purpose manifold-aware architectures have emerged, such as Riemannian residual neural networks (Katsman et al., 2023), which extend ResNets to arbitrary manifolds using exponential maps and vector fields.

