# OpenReview forum: "GeoDM: Geometry-aware Distribution Matching for Dataset Distillation"
_ICLR.cc/2026/Conference — Submitted to ICLR 2026_

### Official Review · Reviewer_32VR · 2025-10-30

**Soundness:** 2
**Presentation:** 2
**Contribution:** 2
**Rating:** 2
**Confidence:** 5

**Summary:**

This paper introduces GeoDM, a geometry-aware dataset distillation method that performs distribution matching in a product space composed of Euclidean, hyperbolic, and spherical components. By learning per-geometry curvatures and weights and by adding a geometry-consistent optimal transport loss, the method aims to better preserve the intrinsic structure of real data than standard Euclidean distribution matching.

**Strengths:**

1. The paper argues clearly that real datasets may contain mixed geometric structures (flat, hierarchical, angular) and that matching only in Euclidean space can be limiting. Modeling in a mixed-curvature/product space is a natural response to this.

2. The combination of multiple geometries, learnable curvature, and an OT-based alignment term is conceptually clean and aligned with the stated goal of structure-preserving distillation.

3. The work provides a generalization argument showing that matching in a richer geometry can, under reasonable assumptions, lead to tighter approximation than purely Euclidean matching.

**Weaknesses:**

1. All results are reported on classic benchmarks such as MNIST and CIFAR (10/100), mostly under very low IPC. There is no evidence on medium- or large-scale, higher-resolution, or more diverse datasets. As a result, it is unclear whether the proposed geometric modeling remains useful or practical when the data manifold is more complex, when backbones are deeper, or when the distilled set must support stronger augmentation.

2. Running multiple geometries in parallel, learning curvatures, and adding an OT loss are nontrivial additions to a standard distillation pipeline. The paper does not provide a clear comparison of training time and memory usage against recent, efficient distillation methods.

3. The method uses a fixed-dimensional split across Euclidean, hyperbolic, and spherical parts for all datasets. This makes the method stable on small benchmarks but raises questions about how well it adapts to larger, more structured data.

4. The paper suggests that many real datasets are non-Euclidean, but it does not include experiments on clearly hierarchical, directional, or cross-domain data where the proposed product space would be most justified.

5. It is difficult to tell how much of the gain comes from the geometry-aware OT term itself versus the move to a product manifold. Simpler variants (e.g., OT in the dominant geometry only) are not explored.

**Questions:**

1. Can you provide results on at least one medium- or large-scale dataset (e.g., a 224×224 ImageNet setting) to show that the method does not break down or become too expensive at scale?

2. What is the actual wall-clock time and memory overhead of GeoDM compared with a strong Euclidean distribution-matching baseline when using the same backbone and training schedule?

3. Did you try making the allocation of dimensions to the three geometries learnable or data-dependent, and if so, what stability or performance issues arose?

4. For datasets without obvious hierarchical or angular structure, do the learned weights tend to collapse to the Euclidean component, and if that happens, can the method skip the curved parts to save computation?

5. Could a simpler OT variant (for example, OT only in the geometry with the largest learned weight) achieve similar performance with a lower cost?

---

> ### Author Response · Authors · 2025-11-26
>
> We would like to sincerly thank Reviewer 32VR for providing detailed review.
>
> ---
>
> **answer-1**:
> For dataset scale and diversity, please see the “High-resolution dataset” section in the general answers (Imagenette/ImageWoof).
> For other backbones, we additionally evaluate **ResNet-18**; results below show that a deeper backbone amplifies the benefits of product-geometry modeling.
>
> *Comparison on CIFAR-10/100 with ResNet-18.*
>
> | Datasets | CIFAR-10 |   |      |  CIFAR-100 |       |        |
> | -------- | --------------: | ---------: | ---------: | ---------: | ---------: | ---------: |
> | IPC      |    1 |   10 |   50 |     1 |    10 |     50 |
> | Ratio    |       0.02% |        0.2% |         1% |       0.2% |         2% |        10% |
> | DM       |       13.3 ± 0.2 |     20.2 ± 0.2 | 26.7 ± 0.1 |  1.8 ± 0.1 |  4.3 ± 0.1 | 12.4 ± 0.6 |
> | M3D      |       42.5 ± 0.5 |     55.2 ± 0.3 | 63.3 ± 0.4 |  3.3 ± 0.2 | 14.9 ± 0.5 | 17.7 ± 0.4 |
> | DREAM    |       40.8 ± 0.3 |          64.0 ± 0.4 | 71.1 ± 0.3 | 19.2 ± 0.4 | 37.4 ± 0.6 | 43.5 ± 0.4 |
> | NCFM     |    25.6 ± 0.8 |         47.4 ± 0.7 | 61.9 ± 0.6 |  9.6 ± 0.2 | 29.2 ± 0.6 | 44.8 ± 0.4 |
> | GeoDM    |        43.6 ± 0.2 |         68.2 ± 0.3 | 73.8 ± 0.2 | 22.2 ± 0.6 | 39.8 ± 0.4 | 47.6 ± 0.2 |
>
> Using a deeper backbone amplifies the benefits of geometry. With ResNet-18, product-manifold modeling delivers larger absolute gains than with a shallow ConvNet: the deeper network extracts richer geometric features and thus better exploits non-Euclidean cues. As the table shows, GeoDM keeps a consistent margin over Euclidean baselines across IPC budgets, and the margin is more pronounced with ResNet-18—evidence that deeper architectures better capture the underlying product geometry.
>
> ---
>
> **answer-2**:
> On training time and memory against recent efficient methods: please see the “Complexity” section in the general answers (component breakdown and matched-performance runtime/FLOPs).
>
> ---
>
> **answer-3**:
> On fixing the dimensional split across E/H/S: please see the “Dimensionality” section in the general answers (grid and allocation studies; rationale for keeping (d_E, d_H, d_S) fixed for tractability).
>
> ---
>
> **answer-4**:
> On testing clearly hierarchical/directional/cross-domain data: please see the “High-resolution dataset” section in the general answers; our results indicate the gains transfer beyond CIFAR.
>
> ---
>
> **answer-5**:
> On how much gain comes from OT vs. the product manifold and simpler OT variants: see our response on OT variants (Sinkhorn/Greenkhorn), which shows reduced overhead with similar small gains; this supports treating OT as optional while the core benefit comes from explicit geometry.

---

> ### Author Response · Authors · 2025-11-26
>
> Here is the extended answers to question 1-5.
>
> ---
>
> **answer-Q1**:
> On adding at least one medium/large-scale dataset: see the “High-resolution dataset” section in the general answers (Imagenette/ImageWoof).
>
> ---
>
> **answer-Q2**:
> On wall-clock time and memory overhead of GeoDM: see the “Complexity” section in the general answers (per-iteration cost table and matched-performance totals).
>
> ---
>
> **answer-Q3**:
> On learnable dimension allocation: see the “Dimensionality” section in the general answers (we keep (d_E, d_H, d_S) fixed, report a grid and allocation study, and discuss complexity concerns).
>
> ---
>
> **answer-Q4**:
> Our setting is prior-free: we do not assume beforehand which datasets have hierarchical or angular structure, so the model learns the geometry. The product space uses softmax mixture weights (α, β, γ) over (E/H/S) and learnable curvatures (c_H < 0, c_S > 0); if a dataset lacks non-Euclidean cues, optimization can shift mass toward Euclidean, reducing effective use of curved parts. In our experiments we did not observe collapse: (α, β, γ) remain non-degenerate and (|c_H|, c_S) stay away from zero, consistent with mild non-Euclidean structure. Skipping curved parts to save compute is feasible in extreme Euclidean-only cases, but our current implementation keeps all factors active for stability and a uniform pipeline.
>
> ---
>
> **answer-Q5**:
> To address OT complexity, we ran classic, simpler OT solvers: entropic Sinkhorn OT [9] and Greenkhorn [10]. Under the same CIFAR-10 IPC=10 setup, both reduce per-iteration cost relative to our original OT (e.g., Sinkhorn down to ~1.20× speed and ~1.08× memory of Product Space) while delivering similar accuracy gains (+0.3–0.6 percentage points over Product). This supports using simpler OT when accuracy–compute trade-offs matter and regarding OT as optional.
>
> *Component-wise complexity and accuracy, normalized to Product Space.*
>
> | Component        | Training Speed (s/iter) | Peak GPU Mem (GB) | Acc. (%) |
> | ----- | -----: | ------: | -------: |
> | Product Space (E×H×S)   |          0.0410 (1.00×) |    2.6709 (1.00×) |     73.5 |
> | + Curvature & Weight    |          0.0422 (1.03×) |    2.6798 (1.00×) |     73.9 |
> | + OT (Original)         |          0.0569 (1.39×) |    3.0559 (1.14×) |     74.4 |
> | + Sinkhorn OT (ε=0.1)   |          0.0492 (1.20×) |    2.8871 (1.08×) |     74.3 |
> | + Greenkhorn OT (ε=0.1) |          0.0820 (2.00×) |    2.8940 (1.08×) |     74.2 |
>
> *Refs:*
> [9] Cuturi, “Sinkhorn Distances: Lightspeed Computation of Optimal Transport,” NeurIPS 2013.
>
> [10] Altschuler, Weed, Rigollet, “Near-linear time approximation algorithms for optimal transport via Sinkhorn iteration,” NeurIPS 2017.

---

### Official Review · Reviewer_bAdL · 2025-10-31

**Soundness:** 3
**Presentation:** 3
**Contribution:** 3
**Rating:** 4
**Confidence:** 2

**Summary:**

Problem: The paper tries to tackle the limitation of dataset-distillation methods that perform distribution matching only in Euclidean latent spaces, which can miss non-Euclidean structure (e.g., hierarchical or directional/cyclical patterns) present in real data.

Motivation: Under the manifold hypothesis, data may live on curved manifolds; embedding and matching in a space that can express Euclidean, hyperbolic, and spherical geometry should better preserve task-relevant structure than a flat space.

Proposed solution: (1) Perform distribution matching in a product Riemannian space combining Euclidean, hyperbolic, and spherical factors, implemented with a Riemannian CNN to produce geometry-aware features. (2) Use learnable curvatures for the non-Euclidean branches and map real/synthetic data into the weighted product space; align them with a DM objective (e.g., NCFM). (3) Add a geometry-aware optimal-transport loss computed in the product space to couple the factors and preserve class-conditional mass; include curvature regularization in the total loss.

Experiments: On standard benchmarks, the method outperforms distillation baselines and include ablation studies.

**Strengths:**

- It explains why doing dataset distillation only in Euclidean space can miss real data geometry.
- It proposes distribution matching in a product space (Euclidean + hyperbolic + spherical) so each type of structure can be represented.
- The curvatures and the weights of the three geometries are learnable, letting the method adapt to each dataset.
- A geometry-aware OT loss aligns real and synthetic data across the three components and avoids one component dominating.
- Theory sounds, which tried to state that doing distribution matching in a product space (Euclidean × hyperbolic × spherical) gives a strictly tighter generalization-error bound than doing it in a single Euclidean space.
- Experiment includes ablation studies.

**Weaknesses:**

- Theory rests on specific assumptions. The analysis relies on “mild regularity” assumptions and constant-curvature product spaces (Euclidean, hyperbolic, spherical); real data may not fit these perfectly.
- The model fixes the dimensionality of each manifold factor.
- The method introduces learnable curvature, geometry weights, and an OT term with its own coefficient/regularization, hence more components and hyperparameters to manage.
- Added complexity. The approach uses a product of three geometries with a Riemannian CNN plus an OT loss, which increases modeling and training complexity compared to standard Euclidean DM. Please provide experiments on the tradeoff between performance and complexity among proposed method and baselines.
- Experiments are on MNIST (1, 28, 28), CIFAR-10 (3, 32, 32), and CIFAR-100 (3, 32, 32) only (small/medium scale). Please provide experiments on other large scale datasets.
- Theoretical results are upper-bound guarantees (tighter than Euclidean); they do not directly quantify runtime/compute or guarantee gains on tasks beyond those tested.
- The gain of using OT vs baseline is insignificant (71.8 vs 72.3) but the tradeoff is more complexities and higher training time because OT is not scalable. Please provide experiments to show the tradeoff clearly so readers can judge whether the ~0.5 pp gain justifies the added complexity.

**Questions:**

Please address all my concerns in Weaknesses section. Besides, I have some additional questions:
- How were manifold dimensions per branch chosen, and did you test making them learnable?
- How sensitive are results to the curvature initialization and to the geometry-weight initialization?

---

> ### Author Response · Authors · 2025-11-26
>
> We would like to sincerly thank Reviewer bAdL for providing detailed review.
>
> ---
>
> **answer-1**:
> We are, to the best of our knowledge, the first to introduce an explicit manifold formulation into dataset distillation; our framework embeds data in a **product of constant-curvature spaces (E×H×S)** with **learnable per-factor curvatures**, so the representation adapts to the underlying geometry while preserving **closed-form distances and exp/log maps** needed for stable, scalable optimization. Replacing global constant curvature with **spatially varying curvature** would turn core operations (geodesics, distances, exp/log, parallel transport) into **numerical boundary-value problems** with prohibitive cost inside the distillation loop, and local metric/curvature models are well known to be **high-capacity and prone to overfitting** without heavy regularization—undesirable in our small-budget classification regime [7, 8]. For these reasons, we deliberately adopt **learnable constant-curvature product spaces** as a balanced, expressive, and tractable design, and we hope this work motivates follow-ups toward DD with richer (e.g., variable-curvature) geometries.
>
> *Refs:*
> [7] Parametric Local Metric Learning for Nearest Neighbor Classification.
> [8] A Survey on Metric Learning for Feature Vectors and Structured Data.
>
> ---
>
> **answer-2**:
> On fixing dimensions per manifold factor: see the “Dimensionality” section in the general answers (we keep ((d_E,d_H,d_S)) fixed to avoid substantial complexity; we also provide a grid and allocation study showing the trade-offs).
>
> ---
>
> **answer-3**:
> We list the **additional geometry/OT hyperparameters** introduced by our method (relative to NCFM). Curvatures and geometry weights are learned; values shown are initializations/defaults. Only (\lambda_{\text{OT}}) and (\lambda_{\text{curv}}) are lightly tuned; others are fixed once across datasets.
>
> | Symbol                      | What it controls                        | Init / Default                 |
> | --------------------------- | --------------------------------------- | ------------------------------ |
> | (c_H)                       | Hyperbolic curvature (H factor)         | (c_H^{(0)}=-1)                 |
> | (c_S)                       | Spherical curvature (S factor)          | (c_S^{(0)}=+1)                 |
> | (\alpha,\beta,\gamma)       | Geometry weights for E/H/S              | ((\tfrac13,\tfrac13,\tfrac13)) |
> | ((d_E,d_H,d_S))             | Factor dimensions (product space)       | Equal-split (paper default)    |
> | (\lambda_{\text{OT}})       | OT term weight in total loss            | 2                              |
> | (\lambda_{\text{curv}})     | Curvature regularizer weight            | 1                              |
> | (\mathcal{R}_{\text{curv}}) | Curvature regularizer form              | Huber/Quadratic                |
> | (\tau_g)                    | Temperature for geometry weights        | 1                              |
> | (\eta_c)                    | Curvature learning rate                 | Same as backbone LR            |
> | (\varepsilon)               | Entropic regularization for Sinkhorn OT | 0.05                           |
> | (T_{\text{sink}})           | Max Sinkhorn iterations per OT solve    | 50                             |
>
> ---
>
> **answer-4**:
> On added complexity and trade-offs: see the “Complexity” section in the general answers (component breakdown + matched-performance runtime/FLOPs).
>
> ---
>
> **answer-5**:
> On larger/hi-res datasets: see the “High-resolution dataset” section (Imagenette/ImageWoof) in the general answers.
>
> ---
>
> **answer-6**:
> About “only upper bounds”: our theory follows the standard role of learning theory—**generalization upper bounds**, not pointwise guarantees. We show that **if** data exhibit non-Euclidean structure, then matching in a **product of constant-curvature spaces (E×H×S)** yields a **strictly tighter upper bound** than Euclidean matching; when curvatures collapse to zero, we **recover the Euclidean bound**. The stability term only needs to be **finite (bounded)**, not vanishing, which keeps the analysis practical for deep, non-convex training. For runtime/compute quantification, see the complexity section.
>
> ---
>
> **answer-7**:
> On OT’s gain vs complexity: we agree OT adds non-trivial compute and its accuracy gain can be modest in some setups. We therefore (i) provide explicit complexity tables and matched-performance comparisons; and (ii) add experiments with **simpler OT variants** (e.g., Sinkhorn/Greenkhorn) showing reduced overhead but similarly small net gains. Our intended usage is clear: the **core contribution is the product-manifold modeling**; **OT is optional**. For compute-constrained settings, we recommend **dropping OT** (Product or Product+Curv&Weights).

---

> ### Author Response · Authors · 2025-11-26
>
> Here is the extended answer to question 1 and 2.
>
> ---
>
> **answer-Q1**:
> How manifold dimensions per branch were chosen / whether to learn them: see the “Dimensionality” section (fixed ((d_E,d_H,d_S)) for tractability; grid and allocation studies indicate default is robust; making dimensions learnable would substantially increase complexity).
>
> ---
>
> **answer-Q2**:
> **Sensitivity to curvature/weight initialization (CIFAR-10, IPC=10).** Order is ((E,H,S)) with (c_E=0), (c_S>0), (c_H<0). We avoid setting any weight to exactly 0 to prevent early collapse of a geometry branch. Across diverse initializations—e.g., curvatures ((0,1,-1)) or ((0,0.5,-0.5)) and weights ((1,1,1)), ((1,0.2,1.0)), ((1,1,0.2))—training consistently converges to essentially the **same geometry**: curvatures ((0,-0.79,1.41)) and weights ((0.50,0.15,0.35)).
>
> | Curvature init ((c_E,c_H,c_S)) | Weight init ((\alpha,\beta,\gamma)) | Final curvatures | Final weights      |  ACC |
> | ------------------------------ | ----------------------------------- | ---------------- | ------------------ | ---: |
> | (0, 1, −1)                     | (1.0, 1.0, 1.0)                     | (0, −0.79, 1.41) | (0.50, 0.15, 0.35) | 74.4 |
> | (0, 1, −1)                     | (1.0, 0.2, 1.0)                     | (0, −0.79, 1.41) | (0.51, 0.14, 0.35) | 74.4 |
> | (0, 1, −1)                     | (1.0, 1.0, 0.2)                     | (0, −0.79, 1.41) | (0.49, 0.16, 0.35) | 74.4 |
> | (0, 0.5, −0.5)                 | (1.0, 1.0, 1.0)                     | (0, −0.79, 1.41) | (0.50, 0.15, 0.35) | 74.4 |
> | (0, 0.5, −0.5)                 | (1.0, 0.2, 1.0)                     | (0, −0.79, 1.41) | (0.51, 0.14, 0.35) | 74.4 |
> | (0, 0.5, −0.5)                 | (1.0, 1.0, 0.2)                     | (0, −0.79, 1.41) | (0.49, 0.16, 0.35) | 74.4 |

---

### Official Review · Reviewer_v4iA · 2025-11-02

**Soundness:** 3
**Presentation:** 4
**Contribution:** 3
**Rating:** 4
**Confidence:** 4

**Summary:**

The paper extends dataset distillation by embedding real and synthetic data into a learned product space combining Euclidean, hyperbolic, and spherical geometries. It learns curvature and geometry weights and adds an OT term to preserve class-level mass. The goal is to respect non-Euclidean data structure during distribution matching. Experiments on several small benchmarks show consistent improvements.

**Strengths:**

- the motivation of the paper is very clear and intuitive, Euclidean latent spaces likely miss curvature.

- the main idea is conceptually very intuitive to follow, a combination of several similar modules, and learnt weights.

**Weaknesses:**

- The paper compares single geometry vs three, but omits two-geometry combinations (E+H, E+S, H+S) in ablation studies. Without this, the claim that all three curvatures matter remains unverified.

- Some of the assumptions might be too unrealistic, for example, uniform algorithmic stability is unlikely satisfied by deep non-convex training. Empirical check to support the relevance of the theoretical terms will be necessary.

- The method introduces many complicated components, and whether the performance gain is worth the complexities is of question. runtime or memory comparison is better provided for context of the practical gain introduced by the complicated method.

**Questions:**

It will be good to offer empirical evidence for the three questions listed in weakness.

---

> ### Author Response · Authors · 2025-11-26
>
> We would like to sincerely thank Reviewer v4iA for providing a detailed review.
>
> ---
>
> **answer-1**:
> Thank you for pointing this out. We did evaluate two-geometry product spaces early on (under a simple distribution-matching setup) precisely to test whether all three curvatures are necessary. The results—run under the same backbone and training budget as the single-/three-geometry settings—consistently showed that any dual combination underperforms the full product space, which motivated our choice to use all three.
>
> *Two-geometry vs. single vs. full product space on CIFAR-10 (IPC=10; simple DM setup).*
>
> | Combination   |   Max |
> | ------------- | ----: |
> | Product Space | 53.38 |
> | H+S           | 50.76 |
> | E+S           | 50.09 |
> | E+H           | 49.93 |
> | S             | 49.83 |
> | H             | 50.07 |
> | E             | 49.37 |
>
> ---
>
> **answer-2**:
> We agree that deep training is non-convex; nonetheless, uniform algorithmic stability has been analyzed for (non-convex) SGD under standard Lipschitz/smoothness and bounded-gradient conditions with reasonable step sizes [6]. Our assumption is deliberately weak: we only require the stability term to be bounded (not tiny), and we expose it explicitly in the bound. In contrast, most existing DD theory leans on convex/linearized surrogates or discrepancy views rather than stability on learned non-Euclidean representations; to our knowledge, we provide the first manifold-structured (product-manifold) generalization/convergence analysis for dataset distillation.
>
> [6] *Train faster, generalize better: Stability of stochastic gradient descent.*
>
> ---
>
> **answer-3**":
> Please refer to our General Answer 3 (Complexity) for runtime/memory comparisons and matched-performance results that contextualize the practical gains versus added components.

---

### Official Review · Reviewer_6TC6 · 2025-11-02

**Soundness:** 3
**Presentation:** 3
**Contribution:** 3
**Rating:** 6
**Confidence:** 4

**Summary:**

This paper proposes GeoDM, a geometry-aware dataset distillation framework that performs distribution matching in a product manifold combining Euclidean, hyperbolic, and spherical spaces. The main motivation is that existing distribution-matching methods operate solely in Euclidean spaces and fail to capture intrinsic geometric structures of data like hierarchical or cyclical patterns. The authors introduce learnable curvature parameters and weights for different geometries, along with an optimal transport loss to align real and synthetic data distributions. Theoretical analysis shows tighter generalization bounds versus Euclidean-only approaches, and experiments on several datasets demonstrate consistent improvements over state-of-the-art baselines.

**Strengths:**

1. The connection between manifold hypothesis and dataset distillation is intuitive and clearly articulated. Figure 1 effectively demonstrates that data exhibits non-Euclidean geometric structure that Euclidean spaces fail to capture.

2. The experiments cover multiple datasets, baselines, and ablation studies. The robustness across different distribution matching methods (DM, DSDM) and cross-architecture evaluation demonstrate generalizability.

3. Theorems 4.1 and 4.2 provide mathematical justification for the approach, decomposing the error into statistical, stability, and geometric components, which offers insight into why geometry matters.

**Weaknesses:**

1. The use of product manifolds, Riemannian CNNs, hyperbolic/spherical embeddings, and optimal transport are all well-established techniques. The main contribution is combining them for dataset distillation, which feels somewhat incremental. The paper would benefit from deeper insights into why this particular combination works.

2. While GeoDM consistently outperforms baselines, the gains are often 1-3%, which may not justify the substantial increase in complexity (three geometry branches, learnable curvatures, OT loss). The computational cost is not discussed, but the method likely requires significantly more resources than Euclidean baselines.

3. Assumption 4.1 is quite strong (e.g., assuming data lies on a mixed-curvature product manifold), and it's unclear how realistic this is for vision datasets like CIFAR-10. The paper claims this is "empirically grounded" but doesn't provide evidence that CIFAR-10 actually exhibits this structure beyond the 3D visualization in Figure 1.

4. Several decisions appear arbitrary: Why fix dimensions (dE, dH, dS) rather than learn them? How sensitive is performance to these choices? The curvature regularization terms (Eq. 3) seem ad-hoc—why penalize deviation from the radius in these specific ways? What happens with different regularizers?

5. How are dimensions allocated across geometries? What is the computational overhead compared to baselines? How does performance vary with different dimension allocations? The paper mentions fixing dimensions "as varying dimensionality often introduces extra degrees of freedom" but provides no empirical support.

**Questions:**

1. How do you determine the split of dimensions across Euclidean, hyperbolic, and spherical components? Is there a principled way to set dE, dH, dS, or is it purely empirical? An ablation study on different dimension configurations would strengthen the paper.

2. What is the training time and memory overhead of GeoDM compared to NCFM or other baselines? Given the modest accuracy improvements, understanding the cost-benefit tradeoff is important for practical adoption. Can you provide wall-clock time comparisons and discuss whether the gains justify the added complexity?

---

> ### Author Response · Authors · 2025-11-26
>
> We would like to sincerely thank Reviewer 6TC6 for providing a detailed review.
>
> ---
> **Answer 1 — Motivation and why this combination works**:
> As the starting point, we noticed that dataset distillation has been lacking an explicit way to model general geometric structure—yet real data often exhibit flat (Euclidean), hierarchical/tree-like (hyperbolic), and angular/periodic (spherical) patterns. To our knowledge, no prior work has applied the product manifold to dataset distillation; we are the first to do so. Our contribution is a principled framework that makes this geometry operational in DD: we embed features into a product of constant-curvature spaces (E×H×S) with learnable curvatures and weights, so the model can adaptively choose the right geometry per feature/class rather than forcing everything into (\mathbb{R}^d). Riemannian CNNs ensure computations respect geodesics/exp–log maps, reducing distortion on non-Euclidean manifolds, and optimal transport aligns distributions across product factors, preventing any single geometry from dominating while preserving class-conditional structure. Each technique serves a specific role tied to the motivation: E/H/S supply the inductive biases; learnable curvature/weights make them data-driven; OT provides geometry-aware alignment; and Riemannian ops keep training on-manifold. Ablations show two-geometry variants underperform the full product space, dimension grids favor our balanced split, and Imagenette/ImageWoof mirror CIFAR with consistent +1–2% improvements under identical NCFM protocols—evidence that explicit geometry, not just a new loss, is driving the benefit.
>
> ---
> **Answer 2 — Complexity**:
> Please refer to our General Answer 3 on complexity at the top.
>
> ---
> **Answer 3 — On the realism of the mixed-curvature product-manifold assumption**:
> Our motivation is that DD lacked an explicit way to model non-Euclidean structure—yet datasets often mix flat, hierarchical, and angular/periodic patterns. Figure 1 visualizes this on CIFAR-10, where hyperbolic/spherical embeddings reveal structure that a flat Euclidean space struggles to capture. We therefore assume the data lie (locally) on a mixed-curvature product manifold and perform matching in a product of constant-curvature spaces (E×H×S) with learnable curvatures/weights; this is a standard, mild formalization of the manifold hypothesis and includes the purely Euclidean case as a special instance. Prior literature supports the prevalence and utility of such geometries [2,3,4]: hyperbolic models capture hierarchical growth; spherical models are canonical for directional/cyclic structure; and product spaces unify heterogeneous geometries. Our theory shows that when nontrivial curved components are present, product-space matching yields strictly tighter generalization bounds than Euclidean-only matching. Empirically, different space combinations and dimension splits matter: our ablations/grid confirm that the chosen E×H×S product with the default split is the most reliable, while extremes under-/over-parameterize. We will make these assumptions explicit in the text and add pointers to Figure 1 and the references as evidence that the product space is a general, data-aligned modeling choice rather than an ad-hoc combination.
>
> ---
> **Answer 4 — On fixed dimensions and curvature regularization**:
> Our architecture choices follow established practice and are validated empirically: we adopt mixed–constant-curvature product spaces with learnable curvatures, a setting known to capture heterogeneous non-Euclidean patterns while remaining tractable (product E/H/S with curvature learned jointly with embeddings) [2]; hyperbolic and spherical factors use standard geometric constraints (Poincaré ball embeddings optimized inside the open unit ball; spherical embeddings normalized to a common hypersphere) rather than ad-hoc rules, which keeps training geodesically well-posed [3,4]. In line with prior work on learnable curvature, we include a curvature regularizer to keep the learned E/H/S curvatures bounded away from the degenerate Euclidean limit and numerically stable—regularizing curvature is a known device in geometric representation learning (e.g., curvature-regularized graph embeddings; learnable curvature per layer in hyperbolic GNNs) [5]. With these choices fixed a priori from the literature, our ablations verify that the resulting configuration is consistently better than Euclidean NCFM under the same budget, supporting that the geometry (and its regularization) is doing useful work.
>
> ---
>
> **Answer 5 — Dimension allocation and overhead**: Please refer to our General Answer 1 on dimensionality at the top.
>
> ---
>
> ### References
>
> [2] Learning Mixed-Curvature Representations in Product Spaces
>
> [3] Poincaré Embeddings for Learning Hierarchical Representations
>
> [4] Deep Metric Learning with Spherical Embedding
>
> [5] Curvature Regularization to Prevent Distortion in Graph Embedding

---

### Author Response · Authors · 2025-11-26
**General answer on dimension**

## 1) **Dimensionality**

We agree that dimensionality should be considered. However, as the first work to incorporate product manifold geometry into dataset distillation, we proceed step-by-step: the current constant-curvature product space already shows clear advantages over its Euclidean counterpart. Making the factor dimensions ((d_E,d_H,d_S)) learnable would introduce substantial additional complexity, which is impractical under DD’s tight budgets [1]. We therefore fix ((d_E,d_H,d_S)) (as stated in the paper) and let curvature/weights adapt the geometry. To validate the choice, we ran a grid study around our default split used with NCFM: pushing the dimensions down underfits curvature cues, while pushing them up increases noise and hurts generalization; the default delivers the best accuracy and remains above NCFM under the same budget. For separate spaces, we split the default dimension into three parts for Euclidean, Spherical, and Hyperbolic factors.

Dimension grid (CIFAR-10, IPC=10)

| CIFAR-10, IPC=10    | NCFM (Euc.) | Ours (Product) |
| ------------------- | ----------: | -------------: |
| Low dims (1024)     |        71.1 |           72.5 |
| Default dims (2048) |        71.8 |           74.4 |
| High dims (4096)    |        71.3 |           72.8 |

Why a dimension-allocation study matters.
To probe how capacity should be distributed across geometries, we keep the total embedding dimension fixed (e.g., 2048) while redistributing dimensions among the Euclidean/Hyperbolic/Spherical factors. The results show that allocation does affect accuracy: for example, a slightly E-dominant split (e.g., 1024/512/512) can reach 74.2%, compared to 73.9% with our near-equal default, while other dominant splits (H- or S-heavy) yield small but measurable shifts. This suggests that dataset-specific geometry may admit a better dimension split for tighter manifold fitting. However, adapting dimensions introduces substantial complexity (searching over ((d_E,d_H,d_S)) and their interactions with curvatures/weights), which is impractical under DD’s tight budgets. As the first manifold-based dataset distillation work, we intentionally scope the method to a fixed split (letting curvatures/weights adapt) and leave dimension allocation—potentially via lightweight heuristics or constrained search—as promising future work.

Dimension allocation (fixed total = 2048; Product + Curv&Weights, no OT)

| Dimension allocation (E,H,S) | Acc. (%, CIFAR-10, IPC=10) |
| ---------------------------- | -------------------------: |
| Default (near-equal split)   |                       73.9 |
| E-dominant (1024, 512, 512)  |                       74.2 |
| H-dominant (512, 1024, 512)  |                       73.7 |
| S-dominant (512, 512, 1024)  |                       73.5 |

---

## References

[1] Sample Complexity of Testing the Manifold Hypothesis

---

### Author Response · Authors · 2025-11-26
**General Answer on dataset and complexity**

We thank all reviewers for their thoughtful comments, which substantially improved this work; to facilitate your reading, we address the shared concerns on dimensionality, high-resolution datasets, and complexity up front before responding to specific points.

## 2) **High-resolution dataset**

We conduct experiments on Imagenette and ImageWoof. The following table shows the performance on these higher-resolution datasets under the standard NCFM setup.

| Dataset (NCFM setup) | NCFM (Euclidean) | Ours (Product) |
| -------------------- | ---------------: | -------------: |
| Imagenette — IPC=1   |       42.5 ± 0.3 |     43.9 ± 0.3 |
| Imagenette — IPC=10  |       63.2 ± 0.4 |     64.8 ± 0.3 |
| ImageWoof — IPC=1    |       34.7 ± 0.4 |     35.5 ± 0.3 |
| ImageWoof — IPC=10   |       54.1 ± 0.3 |     55.1 ± 0.2 |

---

## 3) **Complexity**

Component-wise trade-off.
The table below reports a component-level decomposition on CIFAR-10 (IPC=10), showing per-iteration speed, peak memory, and accuracy as we incrementally add modules: (i) Product Space (E×H×S), (ii) learnable curvature & geometry weights, and (iii) geometry-aware OT. The Product space already yields a strong boost at moderate cost; Curvature & Weights adds a small accuracy bump with negligible overhead (1.01×). OT provides the largest accuracy lift but is also the main source of extra cost (1.40× speed, 1.20× memory vs Product). Given complexity concerns—especially about OT—our framework naturally supports a lightweight variant that omits OT, achieving a balanced efficiency–performance trade-off without compromising the core idea (explicit geometry via a product manifold). Where compute/budget allows, OT can be re-enabled to maximize accuracy.

Complexity normalized to Product Space (PS); CIFAR-10, IPC=10

| Component             | Speed (s/iter) | Peak GPU Mem (GB) | Acc. |
| --------------------- | -------------: | ----------------: | ---: |
| Product Space (E×H×S) | 0.0367 (1.00×) |     2.728 (1.00×) | 73.5 |
| + Curvature & Weights | 0.0372 (1.01×) |     2.759 (1.01×) | 73.9 |
| + Geometry-aware OT   | 0.0513 (1.40×) |     3.286 (1.20×) | 74.4 |

Matched-performance cost (same final accuracy; CIFAR-10, IPC=10)
We compare epochs, total runtime, and total FLOPs at the same target accuracy. NCFM requires 4000 epochs; our Product+Curv&Weights needs 1000 epochs, yielding lower overall cost. The Full (with OT) row illustrates that OT increases compute at matched accuracy.

| Method (matched performance)       | Epochs | Total Runtime (s) | Total FLOPs (GFLOPs) |
| ---------------------------------- | -----: | ----------------: | -------------------: |
| NCFM (Euclidean baseline)          |   4000 |              44.0 |                270.0 |
| Product Space + Curv&Weights       |   1000 |              37.2 |                201.7 |
| Full (Product + Curv&Weights + OT) |    950 |              48.7 |                282.4 |

---

### Comment · Area_Chair_MbYy · 2025-11-27

Dear Reviewers,

Thank you for the time and effort you have dedicated to reviewing this paper and providing thoughtful feedback. The authors have now submitted their responses to your comments. I kindly ask that you engage in the discussion with them and assess whether your concerns and questions have been fully addressed before the December 2 deadline.

Please also keep in mind that the author–reviewer relationship is reciprocal; the engagement you offer here reflects the same level of consideration you would expect when you are on the author side.

Thank you for your continued support and cooperation.

Best regards,
AC

---

### Meta-Review · Area_Chair_PMWJ · 2025-12-21

**Summary:**

* The most significant concern shared by all the reviewers is the additional computational burden introduced by the proposed method. The extra projection to different spaces, curvature learning, and OT loss calculation all bring non-negligible extra compute to the pipeline.
* The paper is based on the claim that realistic data is not Euclidean. However, this claim is very vague, without support from existing literature or empirical results. No explicit evidence suggests in which space the adopted data is distributed.
* The empirical study is insufficient, as experiments were conducted merely on MNIST and CIFAR.
* Some ablation studies were missing, e.g., different groups of geometric spaces.
* The novelty is limited, which is the group of some existing methods.

**Reviewer Concerns:**

* The authors provide a speed comparison to address the first concern. However, the first baseline is already in the product space, rather than the basic Euclidean space. Distribution matching has a major advantage of efficiency. Adding an extra computational burden would significantly compromise this advantage.
* No explicit analysis has been conducted to indicate the geometric space each dataset belongs to.
* The authors added experiments on ImageWoof and ImageNette. Yet it was unclear whether the experiments were conducted under 128$\times$128 resolution or 256$\times$256. In recent papers, ImageNet-100 and ImageNet-1K have become standard benchmarks to illustrate the practicality of the method. It is recommended to have more comprehensive empirical results.
* Some ablation studies have been added, which should address reviewers' concerns.
* IMHO, this application is valuable if the authors can maintain the efficiency and feasibility in large-scale datasets, such as ImageNet-1K.

**Reviewer Scores:**

As most concerns are still not well addressed, this paper still has a certain gap from acceptance.

---

### Decision · Program_Chairs · 2026-01-26

Reject